# MukB ATPases are regulated independently by the N- and C-terminal domains of MukF kleisin

Katarzyna Zawadzka[†], Pawel Zawadzki[‡], Rachel Baker, Karthik V Rajasekar, Florence Wagner, David J Sherratt*, Lidia K Arciszewska*

Department of Biochemistry, University of Oxford, Oxford, United Kingdom

*For correspondence:
david.sherratt@bioch.ox.ac.uk
(DJS);
lidia.arciszewska@bioch.ox.ac.uk
(LKA)

Present address: [†]Department of Biology, Adam Mickiewicz University, Poznan, Poland; [‡]Division of Molecular Biophysics, Adam Mickiewicz University, Poznan, Poland

**Abstract** The *Escherichia coli* SMC complex, MukBEF, acts in chromosome segregation. MukBEF shares the distinctive architecture of other SMC complexes, with one prominent difference; unlike other kleisins, MukF forms dimers through its N-terminal domain. We show that a 4-helix bundle adjacent to the MukF dimerisation domain interacts functionally with the MukB coiled-coiled 'neck' adjacent to the ATPase head. We propose that this interaction leads to an asymmetric tripartite complex, as in other SMC complexes. Since MukF dimerisation is preserved during this interaction, MukF directs the formation of dimer of dimer MukBEF complexes, observed previously in vivo. The MukF N- and C-terminal domains stimulate MukB ATPase independently and additively. We demonstrate that impairment of the MukF interaction with MukB in vivo leads to ATP hydrolysis-dependent release of MukBEF complexes from chromosomes.
DOI: https://doi.org/10.7554/eLife.31522.001

## Introduction

SMC (<u>S</u>tructural <u>M</u>aintenance of <u>C</u>hromosomes) complexes have important roles in managing and processing chromosomes in all domains of life (*Gligoris and Löwe, 2016*; *Nolivos and Sherratt, 2014*; *Uhlmann, 2016*). The distinctive architecture of SMC proteins is conserved with the N-and C-terminal globular domains coming together to form an ATPase head and the intervening polypeptide folding upon itself to form ~50 nm long intramolecular coiled-coil arms, with a dimerisation hinge distal from the head (*Figure 1A*). Upon ATP binding, the heads of SMC dimers engage to generate two ATPase active sites (*Haering et al., 2002*; *Lammens et al., 2004*). In eukaryotes, SMC complexes are exclusively heterodimeric, whilst those in bacteria are homodimers. Nevertheless, the distinctive SMC architecture is conserved, with a kleisin protein linking the two ATPase heads of an SMC dimer, thereby forming a large tripartite proteinaceous ring (*Figure 1* inset). Essential accessory 'kite' (<u>k</u>leisin <u>i</u>nteracting winged-helix <u>t</u>andem <u>e</u>lements) or 'hawk' (<u>H</u>EAT repeat subunits containing proteins <u>a</u>ssociated <u>w</u>ith <u>k</u>leisins) bind the kleisin (*Palecek and Gruber, 2015*; *Wells et al., 2017*). Hawk proteins are present in cohesins and condensins, while kites are present in bacterial SMC complexes, including MukBEF, and eukaryote SMC5/6 complexes. This suggests that the bacterial complexes are more evolutionarily related to the SMC5/6 complexes of eukaryotes than to eukaryote cohesins and condensins. A substantial body of work has led to the hypothesis that DNA segments are topologically entrapped within these tripartite rings. ATP binding and hydrolysis are required for the entrapping of DNA within the rings, loading, and for DNA release, unloading (*Arumugam et al., 2003*; *Çamdere et al., 2015*; *Cuylen et al., 2011*; *Gruber et al., 2003*; *Haering et al., 2002*; *Haering et al., 2008*; *Hu et al., 2011*; *Kanno et al., 2015*; ; *Nasmyth, 2011*; *Murayama and Uhlmann, 2014*; *Uhlmann, 2016*; *Wilhelm et al., 2015*).

*E. coli* and its closest γ-proteobacterial relatives, encode an apparently distant SMC relative, Muk-BEF, with little primary sequence homology to other SMCs (*Nolivos and Sherratt, 2014*). Organisms

**eLife digest** Most DNA in a cell is arranged in structures called chromosomes. From bacteria to humans, chromosomes have to be compacted and highly organized to allow the cells to maintain and use their genetic information. In all organisms, large ring-shaped protein complexes play a crucial role in managing chromosomes. They transport and organize DNA thanks to reactions whose precise mechanism remains unknown. In bacteria, MukB and a type of kleisin called MukF are two examples of molecules involved in chromosome management.

Two MukBs join at one end to form a hinge; at the other end, each MukB protein has a neck and a head. The two heads are linked by the kleisin to form a large protein ring, which can open to capture DNA. The MukB heads can trigger a biochemical reaction that creates the energy essential to trap and release DNA during DNA transport.

Here, Zawadzka et al. study how the different components of the MukB-kleisin complex interact with each other to undergo the biochemical reactions that lead to DNA transport. The experiments show that the kleisin joins two MukB heads by attaching the base of one to the neck of the other, asymmetrically closing the ring. The separate interactions of different regions of the kleisin to the head and neck of MukB independently activate the two MukB heads, thereby controlling essential steps in the reactions with DNA. Two MukB-kleisin ring complexes are joined to each other because of a tight interaction between the two kleisin molecules. This leads Zawadzka et al. to suggest that DNA is sequentially grabbed and released from these two rings during DNA transport, similar to how a climbing rope is attached and released through carabiners.

Cells cannot survive or be healthy without their chromosomes being accurately managed. It is still unclear how molecules such as MukBs and kleinsins drive this process. A better picture of their structure and interactions is an essential first step to understand these mechanisms.

DOI: https://doi.org/10.7554/eLife.31522.002

encoding MukBEF have co-evolved a number of other distinctive proteins, some of which interact with MukB physically and/or functionally; specifically, topoisomerase IV and MatP both interact with MukB in vitro and in vivo (*BrezellecBrézellec et al., 2006*; *Hayama and Marians, 2010*; *Hayama et al., 2013*; *Li et al., 2010*; *Nicolas et al., 2014*; *Nolivos et al., 2016*; *Vos et al., 2013*). MukB forms SMC homodimers, whereas MukF is the kleisin and MukE the kite protein that binds MukF (*Palecek and Gruber, 2015*). All three proteins of the MukBEF complex are required for function, and their impairment leads to defects in chromosome segregation, manifested by impairment of segregation of newly replicated origins (*ori*) and mis-orientation of chromosomes with respect to their genetic map within cells (*Danilova et al., 2007*). In rich media, this leads to inviability at higher temperatures and formation of anucleate cells during permissive low-temperature growth (*Niki et al., 1991*; *Yamanaka et al., 1996*).

Where characterised, most SMC complexes bind their cognate monomeric kleisins asymmetrically, with their N-terminal regions binding the SMC 'neck' adjacent to the ATPase head of the molecule distal to the molecule binding the kleisin C-terminus, thereby-forming the tripartite protein ring (*Figure 1* inset) (*Bürmann et al., 2013*; *Gligoris et al., 2014*; *Gligoris and Löwe, 2016*; *Gruber et al., 2003*; *Haering et al., 2004*; *Huis in't Veld et al., 2014*). MukF is an atypical kleisin, in that it forms stable dimers through interacting N-terminal winged-helix domains (WHD), while its C-terminal domain interacts with the MukB head at the cap, as is the case for characterised SMC kleisins (*Fennell-Fezzie et al., 2005*; *Woo et al., 2009*). Therefore, one MukB dimer is expected to bind a MukF dimer and two MukE dimers in the absence of ATP (*Figure 1A*); ATP binding leads to head dimerisation, accompanied by steric expulsion of one of the two MukF C-terminal domains and head engagement (*Woo et al., 2009*, *Figure 1B* left panel).

Here, we reveal that MukF, like other characterised kleisins, interacts functionally with the MukB neck, through a 4-helix bundle in its N-terminal domain, while its C-terminal domain interacts with the MukB head at the cap. We show that this interaction with the MukB neck is required for MukBEF function in vivo, and infer that this interaction is established and broken during cycles of ATP binding and hydrolysis. Impairment of this interaction in vivo leads to ATP hydrolysis-dependent release of MukBEF clusters from chromosomes. Interactions of the MukF N-terminal domain with the MukB

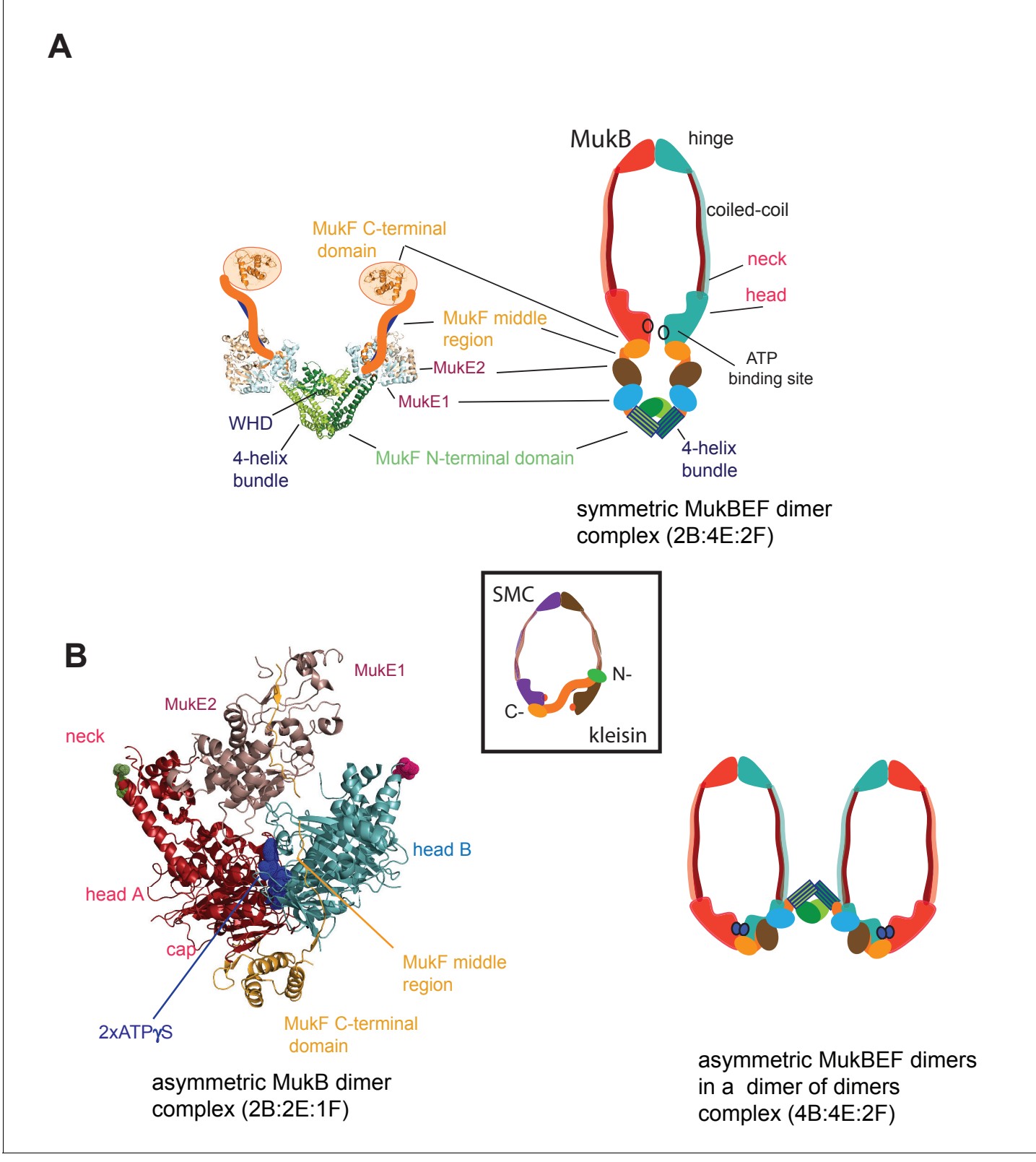

**Figure 1.** MukBEF complexes. (**A**) Left panel; Schematic of a MukF dimer based on the structure of the MukF N-terminal region bound by MukE (pdb, 3EUH; *Woo et al., 2009*) and the C-terminal domain (pdb 3EUK, *Woo et al., 2009*), with a cartoon of the intervening middle region. Right panel; cartoon of a 'classical' view of the proposed symmetric complex of MukBEF in the absence of ATP, with a molecular ratio of 2B:4E:2F. Inset centre; schematic of a typical SMC-kleisin tripartite ring. (**B**) Left panel; crystal structure of *H.ducreyi* hMukE-hMukF(M + C)-hMukBhd$^{EQ}$-ATP-γS asymmetric
*Figure 1 continued on next page*

*Figure 1 continued*

complex (pdb 3EUK, *Woo et al., 2009*). The asymmetric complex is formed by ATP-γS-mediated head engagement; the molecular ratio is 2B:2E:1F; the residues at the coiled-coil exit points are indicated on each head by green and pink dots, respectively. Right panel; cartoon of MukBEF dimer of dimers with stoichiometry of 4B:4E:2F, inferred from in vivo stoichiometry measurements (*Badrinarayanan et al., 2012a*).
DOI: https://doi.org/10.7554/eLife.31522.003

neck and MukF C-terminal domain with the MukB head, activate MukB ATPase independently and additively in vitro, with the addition of both fragments restoring wild type MukB ATPase levels. Each of these ATPase activities was inhibited by MukE, with the inhibition being relieved in the presence of DNA. We show that interaction of the MukF N-terminal domain with the MukB neck did not compromise MukF dimerisation. Therefore, MukF dimerisation in heads-engaged MukBEF complexes directs formation of dimers of MukBEF dimers, thereby explaining the stoichiometry observed in vivo (*Badrinarayanan et al., 2012a*, *Figure 1B* right panel).

## Results

### The MukF N-terminal domain interacts with the MukB neck

Because of the intriguing distinction between dimeric MukF and monomeric kleisins (*Figure 1*), we set out to test, if the MukF N-terminal domain would interact with the MukB neck, thereby exhibiting the architecture of other SMC dimers and their cognate kleisins. In order to undertake an initial characterisation of MukB-MukF interactions, C- and N-terminal MukF Flag-tagged truncations, immobilised on anti-Flag resin, were analysed for binding to intact MukB and its truncated derivatives containing just the ATPase head (MukB$_H$), or the head plus approximately a third of the adjacent coiled-coil region (MukB$_{HN}$). The latter variant would be expected to retain any 'neck' interaction determinants for MukF, based on similarity with kleisin interacting 'necks', adjacent to the SMC heads of other SMC complexes (*Figure 2A*; *Bürmann et al., 2013*; *Gligoris et al., 2014*). MukF C-terminal derivatives, FC1 and FC2, interacted with MukB, and all of its derivatives, as expected, because the MukB ATPase head participates in this interaction (*Figure 2B*; *Figure 1*; *Woo et al., 2009*). FN2, containing the N-terminal dimerisation WHD and an adjacent 4-helix bundle, interacted strongly with intact MukB and MukB$_{HN}$, but not with MukB$_H$, consistent with FN2 interacting with the MukB neck (*Figure 2B*). Since we reproducibly recovered low levels of MukB$_H$ in pulldowns with FN2, the MukB head might also bind FN2 weakly, although we could not substantiate this by further biochemical analyses (below). We detected no interactions of FN1, containing just the WHD involved in MukF dimerisation, with the MukB derivatives. In contrast, FN3, containing the 4-helix bundle (helices 6–9), and FN4, carrying only helices 8 and 9, interacted with MukB and MukB$_{HN}$, but not MukB$_H$ (*Figure 2B*). Consistent with this, FN6 lacking helices 8 and 9 failed to show an interaction in size exclusion chromatography-multi-angle light scattering (SEC-MALS) assays (below) and FN7, lacking helix 9 failed to interact with MukB$_{HN}$ (*Figure 2B* bottom right panel). We conclude that while helices 8 and 9 of the 4-helix bundle are sufficient for interaction with the MukB neck, helix 9 is essential.

To confirm these observations, and to determine the molecular mass of the complexes, we used SEC-MALS (*Figure 3*). MukB$_{HN}$ was monomeric in solution, while FN2 was dimeric, as expected from structural analyses (*Fennell-Fezzie et al., 2005*; *Woo et al., 2009*). When mixed at a molar ratio of 1 MukB$_{HN}$ monomer:1.25 2FN2 (in the figures, we refer to FN2 dimers as 2FN2, to reflect their dimeric state), two additional peaks of masses 165 kDa and 284 kDa were evident in addition to the MukB$_{HN}$ monomers (*Figure 3A* left panel). We interpret these as complexes in which either one or two MukB$_{HN}$ molecules bound independently to a single FN2 dimer. Consistent with this interpretation, more of the larger complexes were observed at higher MukB$_{HN}$ to 2FN2 ratios (3:1; *Figure 3—figure supplement 1*). Therefore, the interaction between the MukF N-terminal domain and the MukB neck does not compromise MukF dimerisation. The relatively low proportion of complexes of stoichiometry MukB$_{HN}$-2FN2-MukB$_{HN}$ as compared to MukB$_{HN}$-2FN2 in the presence of a large excess of MukB$_{HN}$, indicates that binding of the second MukB$_{HN}$ to MukB$_{HN}$-2FN2 complex may be less favourable than binding of the first MukB$_{HN}$ to FN2.

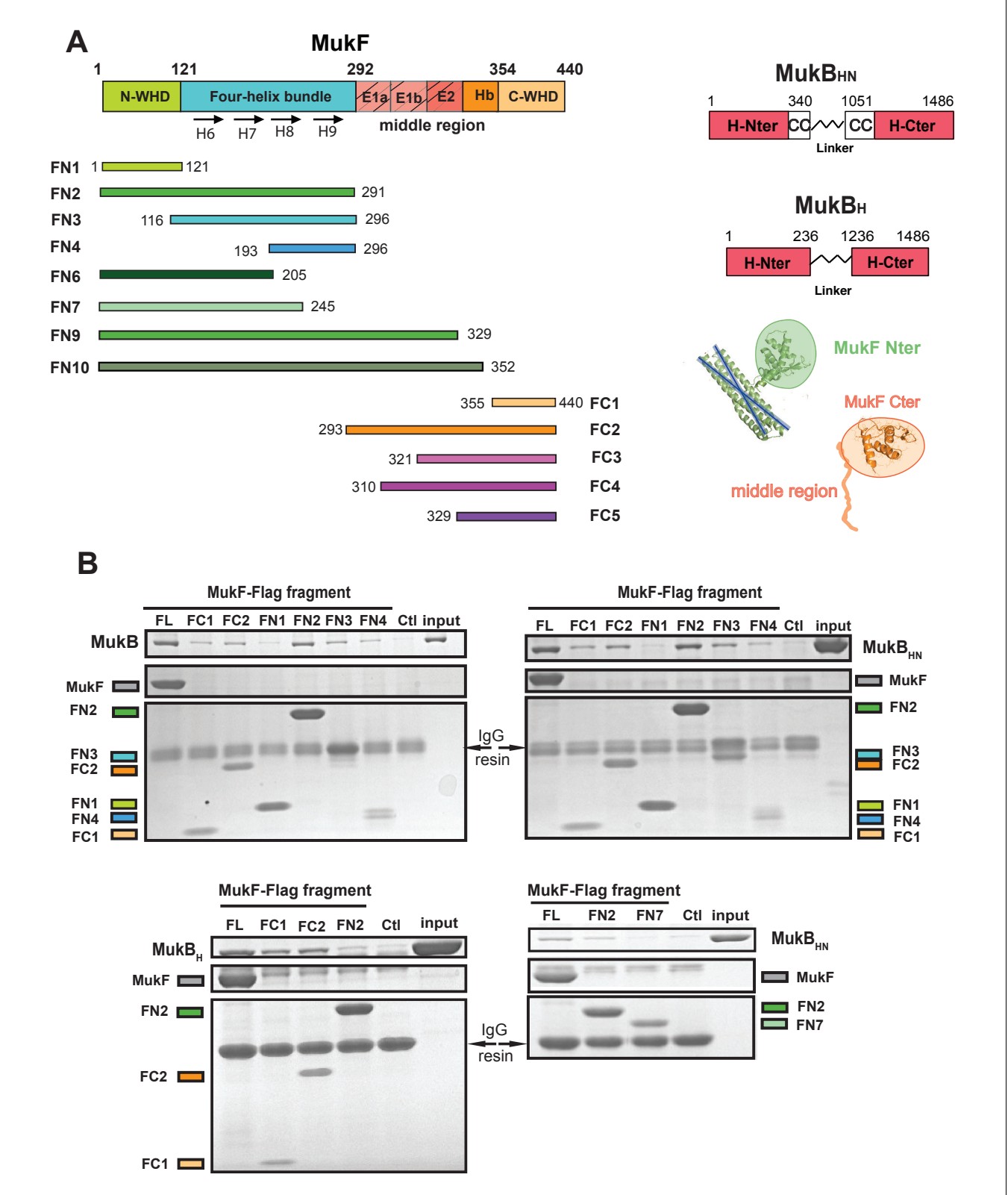

**Figure 2.** The MukF N-terminal domain interacts with MukB neck. (**A**) Left panel; schematics of MukF truncations. The MukF N-terminal WHD is responsible for MukF dimerisation, while the C-terminal WHD interacts with the MukB head (*Fennell-Fezzie et al., 2005*; *Woo et al., 2009*) The middle region contains binding sites for the MukE dimer (E1, E2,) and the C-terminal part of the extended polypeptide that interacts with the MukB engaged head (Hb; *Woo et al., 2009*). Right panel; the MukB head variant (MukB$_H$) carries N- and C-terminal regions that together constitute head domain,

*Figure 2 continued on next page*

*Figure 2 continued*

joined by 18 aa residue flexible linker, while the MukB 'head and neck' variant (MukB$_{HN}$) in addition carries the predicted, head proximal coiled-coil segment (CC) with 104/185 amino acid residues adjacent to the MukB N- and C-terminal domains, respectively (*Li et al., 2009*; *Weitzel et al., 2011*). Cartoons of MukF N- and C-terminal domain structures are included. (B) Pull-down assay using MukF-FLAG tagged fragments as baits for the indicated MukB derivatives. The amounts of recovered MukB, MukB$_{HN}$ or MukB$_H$ are shown within the top boxed portion of the gel in each panel, alongside the MukB derivative input and Ctl., a control with no added bait. Note, that the reduced pull-downs with FN3 and FN4 as compared to FN2, are likely a consequence of reduced concentrations of these baits in extracts.

DOI: https://doi.org/10.7554/eLife.31522.004

In agreement with the Flag-MukF-MukB interaction assays, FC2, but not FN2, formed complexes with MukB$_H$ (*Figure 3A* middle and right panels). FN6, which lacks the two C-terminal helices, 8 and 9, of the 4-helix bundle, failed to form complexes with MukB$_{HN}$ (*Figure 3B*). Addition of ATP did not significantly alter the nature or abundance of complexes containing MukB$_{HN}$ and FN2 or FC2 (*Figure 3—figure supplement 2*). This is consistent with MukB$_{HN}$, which is a monomer in solution, being unable to form stable heads-engaged dimers with either FN2 or FC2 in the presence of ATP.

We next tested whether monomers of MukB$_{HN}$ can simultaneously bind both FN2 and FC2. SEC analysis (*Figure 3C*) showed that mixtures of MukB$_{HN}$, FN2 and FC2 yielded larger complexes (olive green trace) than those formed with MukB$_{HN}$ and FN2 alone (dark green trace), consistent with binding of both FN2 and FC2 to a single monomer of MukB$_{HN}$. Nevertheless, it was not possible to assign precise masses to these by light scattering, because of the dynamic nature of the complexes and an inability to completely resolve them under a range of SEC conditions. Therefore, a complex containing a MukB dimer with unengaged heads, bound to a MukF dimer may be stabilised by MukF interactions to both the MukB head and neck. An equivalent result was observed with *B. subtilis* SMC complexes, with both head and neck of a single SMC molecule being bound simultaneously by kleisin N-and C-terminal domains (*Bürmann et al., 2013*).

To characterise further the interaction of MukF N- and C-terminal domains to MukB, we determined the binding affinities of fluorescently labelled FN2, FN10, FN3 and FC2 using Fluorescence Correlation Spectroscopy (FCS) and Fluorescence Polarization Anisotropy (FPA). Both domains bound to MukB with similar affinities, with K$_d$s in the 9–26 nM range, suggesting that interactions of the N-terminal and C-terminal MukF domains with the MukB neck and head, respectively, are similarly strong (*Figure 3—figure supplement 3*). FN10, which in addition to the N-terminal domain also carries the MukF middle region, bound more tightly to MukB than FN2, consistent with the MukF middle region interacting directly with MukB (*Woo et al., 2009*)

## The MukF C- and N-terminal domains activate MukB ATPase independently and additively

MukB dimers alone had negligible ATPase activity (*Figure 4*), in agreement with previous reports (*Petrushenko et.al., 2005*; *Woo et al., 2009*). Addition of MukF kleisin led to robust MukB ATPase. The steady state ATPase rate was ~21 ATP molecules hydrolysed/min/MukB dimer, under conditions of MukF excess (*Figure 4—figure supplement 1A*). MukF alone did not exhibit ATPase activity. To dissect the MukF requirements for MukB ATPase, we assayed two MukF truncations, containing either the N-terminal domain (FN2), or the C-terminal domain and the middle region (FC2) (*Figure 2A*). Both variants stimulated MukB ATPase (*Figure 4*). Saturating FC2, at a 2.5-fold molar excess, gave 60% of the maximal ATPase obtained with MukF, while saturating FN2 (at a 2.5-fold molar excess) gave 33% of maximum ATPase (*Figure 4—figure supplement 1BC*), while a truncation equivalent to FN2 plus the MukF middle region raised this level to ~50% (later). Addition of FN2 and FC2 together restored ATPase to the level observed with wild type MukF. FN6, lacking helices 8 and 9, did not stimulate MukB ATPase, consistent with its failure to interact with the MukB neck (*Figure 4—figure supplement 1D*). Taken together, the results show that MukB ATPase is activated additively and independently by the N-and C-terminal domains of MukF, with each domain being able to activate ~50% of maximal MukB ATPase.

## Characterisation of the interactions between the MukB neck and MukF

To gain further insight into the interaction of the MukF 4-helix bundle and the MukB neck, variants altered in the MukB neck and MukF helix 9 were analysed for their activity and binding. The

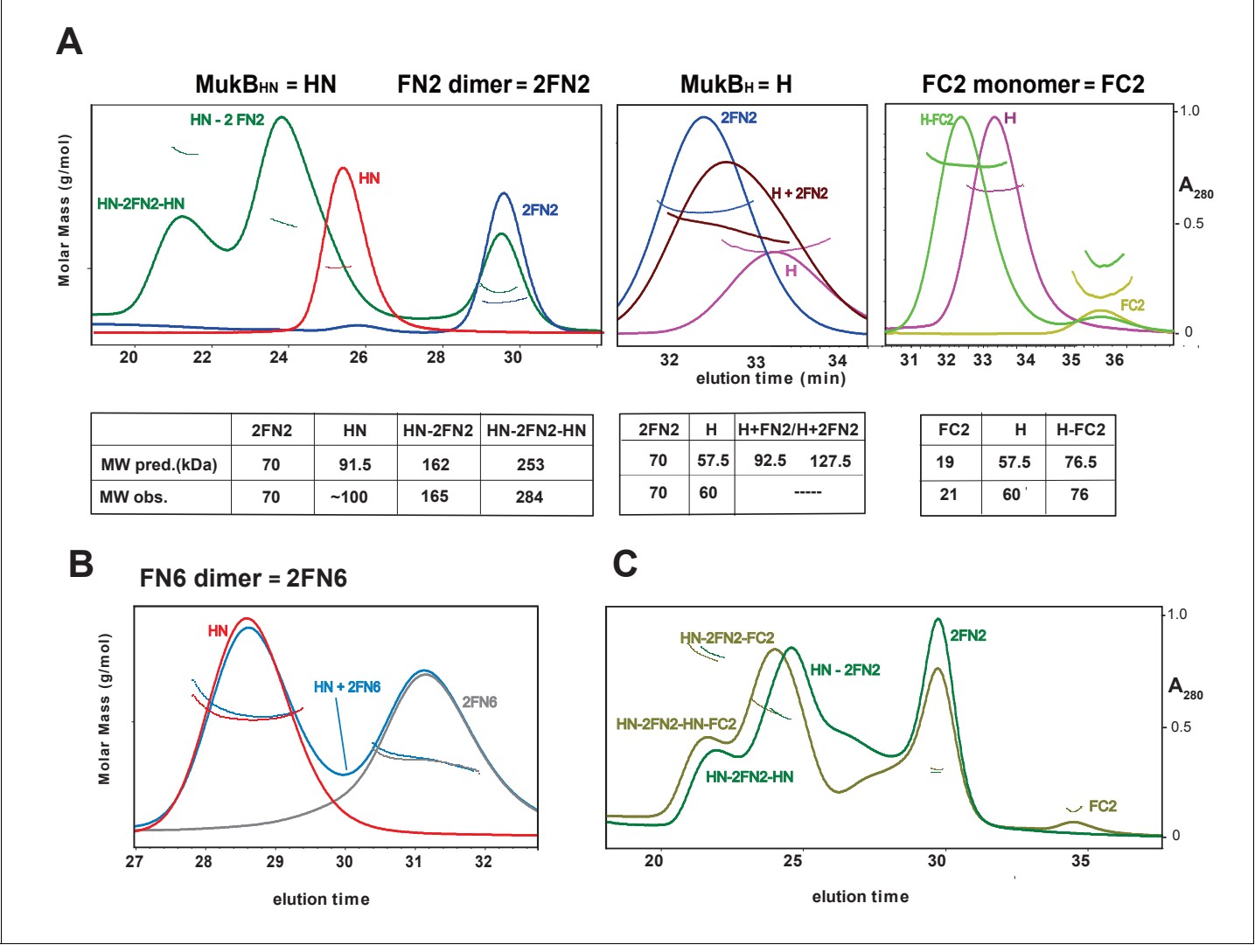

**Figure 3.** Complexes of MukF N- and C-terminal domains with MukB head variants. Binding and stoichiometry of complexes was determined by SEC-MALS. (A) Left panel; MukB$_{HN}$ (red), 2FN2 (blue), and MukB$_{HN}$ + 2FN2 (green) at a 1:1.25 monomer:dimer molar ratio. Middle panel; MukB$_H$ (pink), 2FN2 (blue), and MukB$_H$ + 2FN2 (brown) at 1:0.25 m:d ratio. Right panel; MukB$_H$ (pink), FC2 (lime green), and MukB$_H$ + FC2 (green) at a 1:1 m:m ratio. (B) MukB$_{HN}$ (red), 2FN6 (grey), and MukB$_{HN}$ + 2FN6 (blue) at a 1:0.25 m:d ratio. (C) MukB$_{HN}$ + 2FN2 at a 1:1 m:d ratio (dark green), and MukB$_{HN}$ + 2FN2 + FC2 at a 1:1:1 m:d:m ratio (olive green).

DOI: https://doi.org/10.7554/eLife.31522.005

The following source data and figure supplements are available for figure 3:

**Figure supplement 1.** SEC-MALS analysis of MukB$_{HN}$-2FN2 complexes.
DOI: https://doi.org/10.7554/eLife.31522.006
**Figure supplement 2.** SEC-MALS analysis of MukB$_{HN}$–2FN2 and MukB$_{HN}$–2FN2–FC2 complexes in the absence and presence of ATP (1 mM).
DOI: https://doi.org/10.7554/eLife.31522.007
**Figure supplement 3.** Binding affinities of MukF fragments to MukB.
DOI: https://doi.org/10.7554/eLife.31522.008
**Figure supplement 3—source data 1.** Binding affinities of MukF fragments to MukB.
DOI: https://doi.org/10.7554/eLife.31522.009

mutagenesis strategy was informed by structures of comparable kleisin and SMC neck interactions in yeast cohesin, and *B. subtilis* SMC complexes (*Gligoris et al., 2014*; *Huis in't Veld et al, 2014*, *Bürmann et al., 2013*); see Materials and methods for the mutagenesis strategy). Three variants with triple substitutions in helix 9 of FN2 exhibited an impaired ability to activate MukB ATPase.

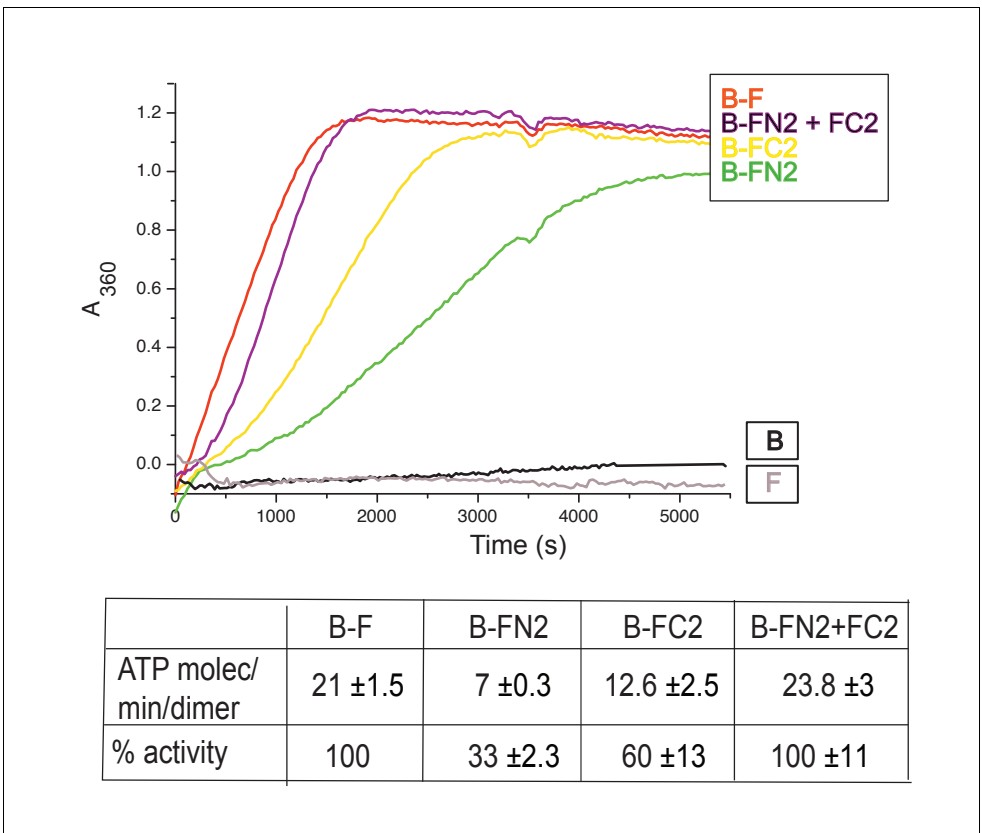

**Figure 4.** MukF N- and C-terminal domains stimulate MukB ATPase. Concentrations in the assays were: MukB, 0.5 µM, MukF/FN2/FC2 1.25 µM, that is, at molar ratio of B:F, 0.5:1.25 monomer equivalent. The curves in the graph represent a single experiment; averages of initial rates ± SD from three experiments are tabulated beneath.
DOI: https://doi.org/10.7554/eLife.31522.010

The following source data and figure supplements are available for figure 4:

**Source data 1.** MukF N- and C-terminal domains stimulate MukB ATPase.
DOI: https://doi.org/10.7554/eLife.31522.013
**Figure supplement 1.** MukF stimulated MukB ATPase.
DOI: https://doi.org/10.7554/eLife.31522.011
**Figure supplement 1—source data 1.** MukF stimulated MukB ATPase.
DOI: https://doi.org/10.7554/eLife.31522.012

FN2m2 (substitutions R279E K283A R286A) displayed a ~10 fold reduction in the ability to activate MukB ATPase, FN2m3 (D261K S265K Q268A), showed a ~2 fold reduction, while FN2m1 (D272K I275K R279D) was reduced by about a third (*Figure 5A*). Moreover, SEC analysis showed that FN2m1 and FN2m2 failed to interact detectably with MukB$_{HN}$ (*Figure 5—figure supplement 1*). Consistent with these results, functional in vivo complementation analysis of the ability of full length MukF variants, containing these sets of mutations, showed that neither MukFm2 nor MukFm3 could complement the temperature-sensitivity of Δ*mukF* cells, while MukFm1 exhibited partial complementation (*Figure 5—figure supplement 2*).

In addition, we analysed MukB variants carrying three double amino acid substitutions in the neck, located near MukB C-terminal head domain. They were designed to be at different locations on the putative candidate coiled-coil helix, that protrudes from the C-terminal subdomain of the MukB head, in positions that were predicted to point towards the MukF 4-helix bundle (*Figure 5B*). MukBm3 (L1219K L1226K) had about 35% of wild type ATPase activity when activated by full length MukF; FN2 was unable to activate its ATPase, while FC2 activated it as efficiently as wild type MukF, consistent with its defect in interaction with FN2 (*Figure 5—figure supplement 3*; *Figure 5—figure supplement 4*). MukBm2 (E1216A E1230A), showed no reduction in ATPase, while MukBm1

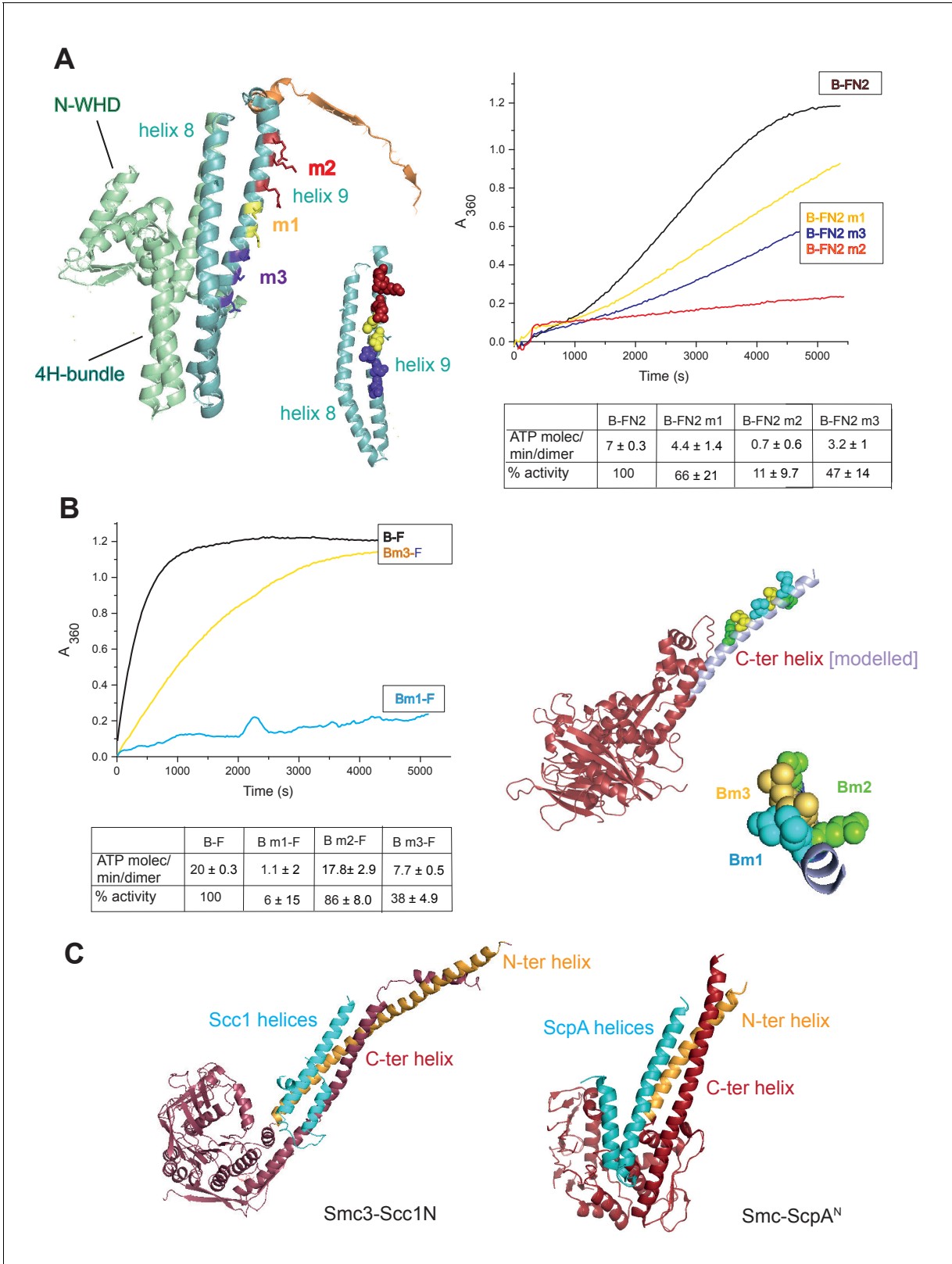

**Figure 5.** Interface between the MukB neck and the MukF four-helix bundle. (**A**) Left panel; cartoon of MukF N-terminal domain fragment carrying the N-terminal dimerisation domain (green) and part of the middle region (orange). Helices 8 and 9 are indicated in cyan. The mutated amino acid residues in variants FN2m1, FN2m2 and FN2m3 are indicated in yellow, red and purple, respectively, (residue R279 was altered in both m1 and m2, but is shown only in m2); views of helices 8 and 9 from different angles are shown separately. Right panel; ATPase activities of the mutated variants; means of initial

*Figure 5 continued on next page*

*Figure 5 continued*

rate measurements from three experiments are tabulated below. (**B**) Left panel; ATPase activities in the presence of MukF, of MukB and MukB variants mutated at the neck, MukBm1, blue, and MukBm3, yellow. Averages of initial rates from three experiments are tabulated underneath. Right panel; monomer of the MukB head (pdb 3EUK, *Woo et al., 2009*); the helix that emerges from the C-terminal subdomain of the head (C-ter helix) and forms the head-adjacent segment of the coiled-coil has been extended by modelling (shown in lilac). Right; enlarged view of the C-ter neck helix from the top with mutated residues shown. (**C**) Interactions of kleisin N-terminal domains with SMC necks. Left panel; Smc3-Scc1N; *Gligoris et al. (2014)*. Right panel; *B. subtilis* SMC-ScpA$^N$; *Bürmann et al. (2013)*. The coiled-coil neck consists of two helical regions protruding from the SMC N-terminal head subdomain (N-ter helix; yellow), and from the C-terminal head subdomain (C-ter helix; red). Kleisin helices are shown in cyan.

DOI: https://doi.org/10.7554/eLife.31522.014

The following source data and figure supplements are available for figure 5:

**Source data 1.** Interface between the MukB neck and the MukF four-helix bundle.
DOI: https://doi.org/10.7554/eLife.31522.023
**Figure supplement 1.** Mutated FN2 fragments were defective in binding to MukB$_{HN}$.
DOI: https://doi.org/10.7554/eLife.31522.015
**Figure supplement 2.** Functional analysis of mutated MukF helix9 variants.
DOI: https://doi.org/10.7554/eLife.31522.016
**Figure supplement 3.** Stimulation of the MukB neck variants, MukBm1 and MukBm3, ATPase by MukF, FN2 and FC2.
DOI: https://doi.org/10.7554/eLife.31522.017
**Figure supplement 3—source data 1.** Stimulation of the MukB neck variants, MukBm1 and MukBm3, ATPase by MukF, FN2 and FC2.
DOI: https://doi.org/10.7554/eLife.31522.018
**Figure supplement 4.** MukBm1 and MukBm3 fail to bind MukF N-terminal fragment.
DOI: https://doi.org/10.7554/eLife.31522.019
**Figure supplement 4—source data 1.** MukBm1 and MukBm3 fail to bind MukF N-terminal fragment.
DOI: https://doi.org/10.7554/eLife.31522.020
**Figure supplement 5.** Functional analysis of mutated MukB neck variants.
DOI: https://doi.org/10.7554/eLife.31522.021
**Figure supplement 6.** Model of the complex made by FN2 dimer binding two monomers of MukB$_{HN}$.
DOI: https://doi.org/10.7554/eLife.31522.022

(M1215K L1222K) had <10% of MukF-stimulated MukB ATPase (*Figure 5B*), indicating that both the MukF N- and C-terminal fragments failed to activate the MukB ATPase of this variant (*Figure 5—figure supplement 3*). Although we believe that this protein can fold correctly, the failure to have its ATPase activated by the MukF C-terminal domain is not yet understood; perhaps substitutions at these residues result in an alteration of the coiled-coil structure adjacent to the head, thereby compromising head engagement. MukBm1 and MukBm3 failed to bind FN2 in FPA assays (*Figure 5—figure supplement 4*).

Consistent with these results, expression in vivo of MukBm1 failed to complement the temperature-sensitive growth defect of Δ*mukB* cells, while MukBm3 exhibited partial complementation. MukBm2 expression fully complemented the Muk⁻ phenotype (*Figure 5—figure supplement 5*). These data suggest that the MukBm3 altered residues (L1219K and L1226K) could either be directly involved in the interaction with MukF 4-helix bundle, or that their replacement by charged lysine residues, interferes with a normal interaction interface.

Using the *E. coli* MukEF crystal structure (pdb, 3EUH; *Woo et al., 2009*), along with the structure of the engaged MukB heads, we modelled a FN2 dimer bound by two monomers of MukB$_{HN}$. This indicated that unless a major conformational change within FN2 dimer takes place upon MukB$_{HN}$ binding, the arrangement of the heads, imposed by interaction of their necks with the 4-helix bundles of the dimer, would be very different from the one revealed by the structure of the engaged MukB heads complex (*Figure 5—figure supplement 6A*). The motifs that compose the two ATPase active sites in each head monomer would be distant and rotated away from each other. Therefore, if simultaneous binding of the two necks within the intact MukB dimer by the two N-terminal domains of MukF dimer is possible, it would produce a complex whose heads would not be able to engage in ATP binding. Whether such a complex is generated at any stage of the MukBEF activity cycle remains to be determined.

In conclusion, the functional interaction between the MukF N-terminal helix 9 and the neck region of MukB coiled-coil revealed and characterised here is equivalent to the similar interaction in other

characterised SMC complexes (*Gligoris et al. (2014)*; *Huis in't Veld et al, 2014*, *Bürmann et al., 2013*).

## MukE inhibits MukBF ATPase

MukE inhibited MukF-stimulated MukB ATPase in steady-state assays (*Figure 6A*; *Bahng et al., 2016*). This inhibition was MukE concentration-dependent (*Figure 6—figure supplement 1*). We then tested whether MukE could equally inhibit the ATPase activated by the isolated C- and N-terminal domains of MukF. The incorporation of MukE into a MukBF complex depends on the asymmetric binding of a MukE dimer to the MukF middle region, which also interacts with MukB head in the engaged MukB heads complex (*Shin et al., 2009*; *Woo et al., 2009*; *Figure 7A* and *Figure 1B* left panel). In the absence of MukE, the N- and C-terminal variants of MukF carrying the entire middle region, FN10 and FC2, respectively, showed 50–60% of wild type MukF activation activity, whereas variants lacking the middle region, FN2 and FC5, showed 25–33% of activation activity, thereby implicating the middle region, whether it be specified by the C- or N-terminal domains, in stabilising or directing, a conformation that optimises ATP hydrolysis. Both the MukF C-terminal head binding fragment (*Figure 7A*, 'Hb') and the MukE binding segment of the MukF middle region (*Figure 7A*, 'E1a, E1b, E2') contributed to the optimal activation activity (*Figure 7B*). The molecular basis underlying the role of this middle region segment in maximising steady state MukB ATPase remains unclear; there are no structural data available to inform how the MukF middle region interacts with the MukB head in the absence of MukE.

MukE inhibited MukB ATPase activated by the MukF N- and C-terminal domain variants that carried complete MukE dimer binding sites (FN9, FN10 and FC2), with a ~4 fold greater inhibition of activation by FN10, as compared to FC2 (*Figure 7B*). MukE was unable to inhibit ATPase stimulated by FN2 and FC5, both of which were lacking MukE binding sites. The effect of MukE on FC4, lacking the N-terminal part of the MukE1 binding site (E1A) was to partially inhibit ATPase activation. These data demonstrate that each of the MukB ATPase activities, stimulated independently by the N- and C-terminal domains of MukF can be inhibited by MukE binding to MukBF.

## DNA binding to MukB relieves MukE-mediated ATPase inhibition

Previous reports have shown no effect of DNA on the ATPase of MukBEF (*Chen et al., 2008*, *Petrushenko et al., 2006*; *Woo et al., 2009*), whereas *B. subtilis* SMC ATPase was reported to be stimulated modestly by DNA (*Hirano and Hirano, 2004*). We confirmed that MukB ATPase is independent of the presence of DNA (*Figure 6B*); addition of 53 bp ds linear DNA at 20-fold excess (10 μM) over MukB (0.5 μM), did not influence MukBF ATPase activity. MukBF ATPase was not dependent on residual DNA contamination of the proteins as judged by the observation that extensive DNase treatment of MukBF did not influence the ATPase level.

DNA alleviated the MukE-mediated inhibition of MukB ATPase. At 5–10-fold excess of DNA over MukB, the ATPase level was restored to ~50% of the level in the absence of MukE (*Figure 6B*). A similar restoration of activity was observed for most other MukF variants, although FC4 exhibited similar MukE-inhibited ATPase activities in the presence and absence of DNA (*Figure 7B*). MukF derivatives, FN2 and FC5, lacking MukE binding sites were not inhibited by MukE and did not respond to DNA.

The position of the DNA binding interface on MukB heads, defined by structure-informed mutational analysis (*Figure 7C*; *Woo et al., 2009*), indicated that DNA binding to this interface could clash with MukE dimer binding to the MukF middle region in a heads-engaged MukBEF complex. Therefore, it seems possible that relief of MukE inhibition by DNA might reflect a competition between MukE and DNA for binding to the MukBF head complex, consistent with the demonstration that MukFE can disrupt MukB-DNA interactions (*Petrushenko et al., 2006b*).

## The MukF N-terminal and C-terminal domains independently modulate MukBEF action in vivo

Since N-terminal and C-terminal domains of MukF could independently and additively bind MukB at the neck and cap, and independently stimulate MukB ATPase activity, we analysed the consequences of disruption of the interactions of endogenous MukF with MukB neck and cap regions in live cells. To this end, we over-expressed either FN2 or FC5 polypeptides from the inducible arabinose

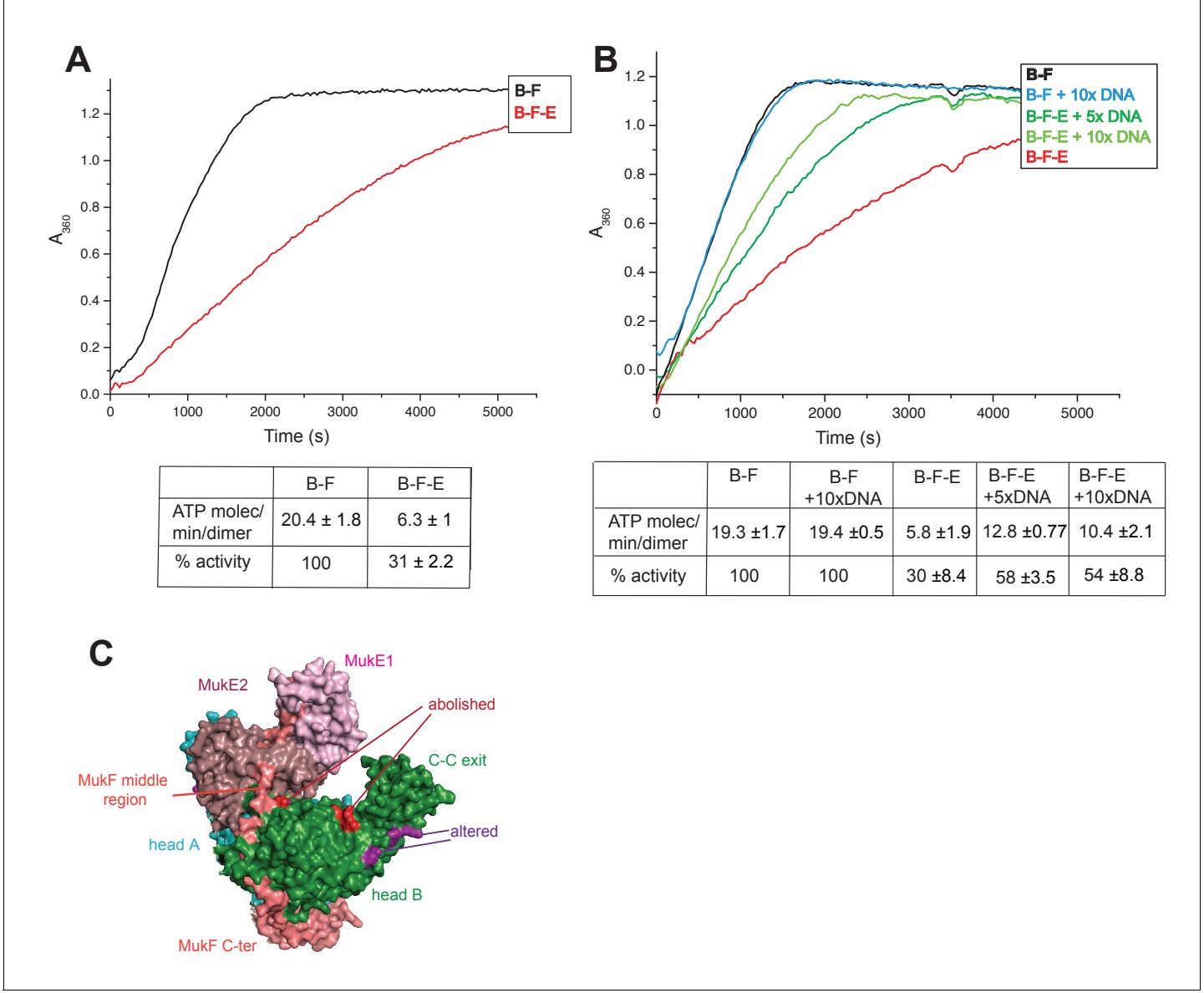

**Figure 6.** Regulation of MukB ATPase. (**A**) MukE inhibits MukBF ATPase. Concentrations: MukB, 0.5 μM, MukF 1.25 μM, and MukE 5.0 μM. (**B**) DNA alleviates MukE-mediated inhibition. ATPase was measured in the presence/absence of 53 bp linear ds DNA fragment at 5x or 10x molar excess over MukB. The average values of the initial rates ± SD from three experiments are tabulated beneath the graphs. (**C**) Surface representation of the MukBEF asymmetric complex with amino acids, whose substitution abolished (red) or altered interactions with DNA (*Woo et al., 2009*).
DOI: https://doi.org/10.7554/eLife.31522.024

The following source data and figure supplements are available for figure 6:

**Source data 1.** Regulation of MukBF ATPase.
DOI: https://doi.org/10.7554/eLife.31522.027
**Figure supplement 1.** Inhibition of MukBF ATPase by MukE.
DOI: https://doi.org/10.7554/eLife.31522.025
**Figure supplement 1—source data 1.** Inhibition of MukBF ATPase by MukE.
DOI: https://doi.org/10.7554/eLife.31522.026

promoter on a multicopy plasmid in cells expressing chromosomal MukBmYPetEF. We assessed MukBEF function by analysing the presence and behaviour MukBmYPetEF clusters observed as fluo-rescent foci associated with the replication origin (*ori*) (*Badrinarayanan et al., 2012a*, *2012b*; *Nolivos et al., 2016*).

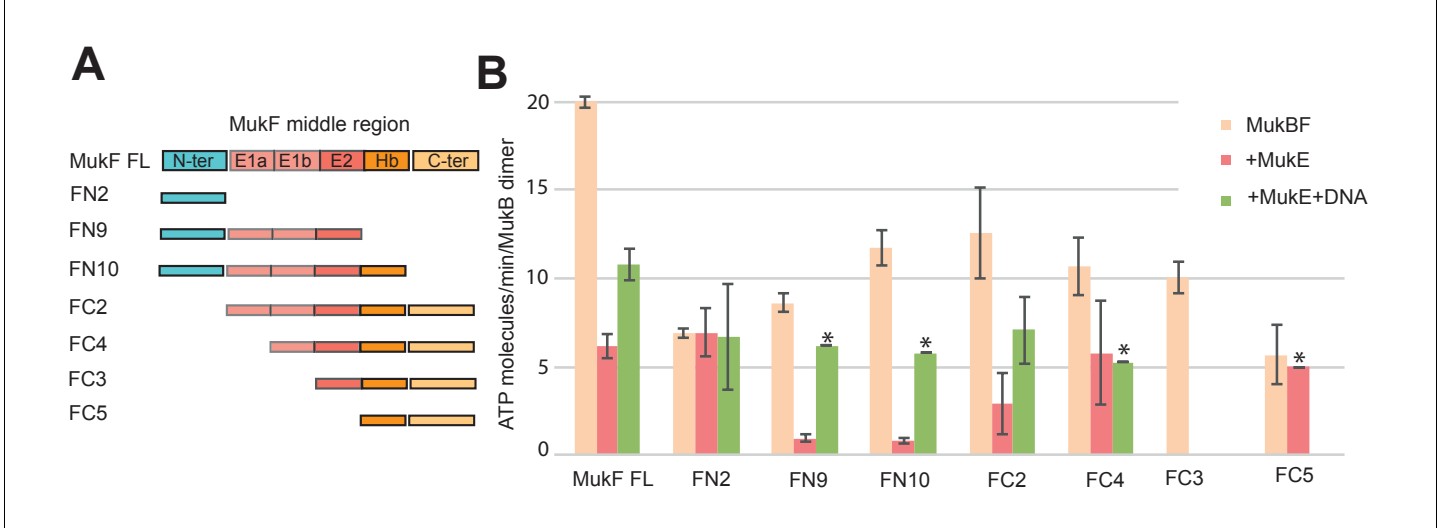

**Figure 7.** Influence of the MukF middle region on the modulation of MukB ATPase by MukE and DNA. (A) One monomer of a MukE dimer binds helical region E1a and part of the acidic linker E1b, while the second MukE monomer binds E2. Hence, MukE binds FC2, containing the entire middle region, but not FN2, which lacks the middle region (*Figure 7—figure supplement 1*). The C-terminal part of the MukF middle region forms an extended polypeptide that binds the MukB head in the asymmetric complex (Hb; *Woo et al., 2009*). (B) Stimulation of MukB ATPase by MukF variants in the presence and absence of MukE and DNA (53nt ds fragment at 10-fold molar excess over MukB). The bars show means of the initial rates ± SD from three independent experiments.

DOI: https://doi.org/10.7554/eLife.31522.028

The following source data and figure supplement are available for figure 7:

**Source data 1.** Influence of the MukF middle region on the modulation of MukB ATPase by MukE and DNA.
DOI: https://doi.org/10.7554/eLife.31522.030
**Figure supplement 1.** SEC-MALS analysis of MukE binding to FC2 and FN2.
DOI: https://doi.org/10.7554/eLife.31522.029

Induced over-expression of FN2 led to a rapid loss of MukBEF foci (half-life of loss ~10 min), whereas FC5 over-expression had a lesser effect on focus loss (half-life of loss ~45 min) (*Figure 8*, *Figure 8—figure supplement 1*). In both cases, residual MukBEF clusters remained *ori*-associated. We then tested if normal cycles of ATP binding and hydrolysis are responsible for the inferred turnover of MukF within functional MukBEF complexes, by testing the effect of fragment production on MukB$^{EQ}$EF complexes that are impaired in ATP hydrolysis and form clusters that turn over very slowly at the replication terminus (*ter*) rather than at *ori* (*Badrinarayanan et al., 2012a*). Over-expression of either FN2 or FC5 had little effect on *ter*-associated fluorescent MukB$^{EQ}$mYPetEF clusters, consistent with the observation in FRAP experiments that there was little turnover of these complexes, presumably as a consequence of their impaired ATP hydrolysis (*Badrinarayanan et al., 2012a*). Nevertheless, we cannot exclude the possibility that the failure to lose MukB$^{EQ}$EF complexes on FN2 or FC5 over-expression is a consequence of the altered cellular localisation of MukB$^{EQ}$EF complexes rather than their impaired ATP hydrolysis. Analysis of the protein composition in MukB$^+$ and MukB$^{EQ}$ extracts, verified comparable high levels of induced expression of FN2 and FC5 in *mukB* and *mukB$^{EQ}$* cells (>100 fold excess over endogenous MukF for both FN2 and FC5, when judged by Western blots; *Figure 8—figure supplement 2*). These observations are consistent with the hypothesis that the MukF interaction with MukB breaks and reforms during cycles of ATP binding and hydrolysis, and that impairment of this interaction leads to loss of functional MukBEF clusters from the chromosome. We have also considered the possibility that the loss of MukBEF clusters from *ori* after over-expression of FN2 results from the disruption of MukF dimers, rather than an opening of a ring interface. If this were the case, we would have expected the same result in *mukB$^{EQ}$* cells.

The relatively low turnover of this interaction as compared to the dwell time of MukBEF complexes in vivo (~50 s) and the rates of ATPase measured in vitro could be a consequence of the chelate effect arising from the fact that when the N- or C-terminal domain is released from the MukB

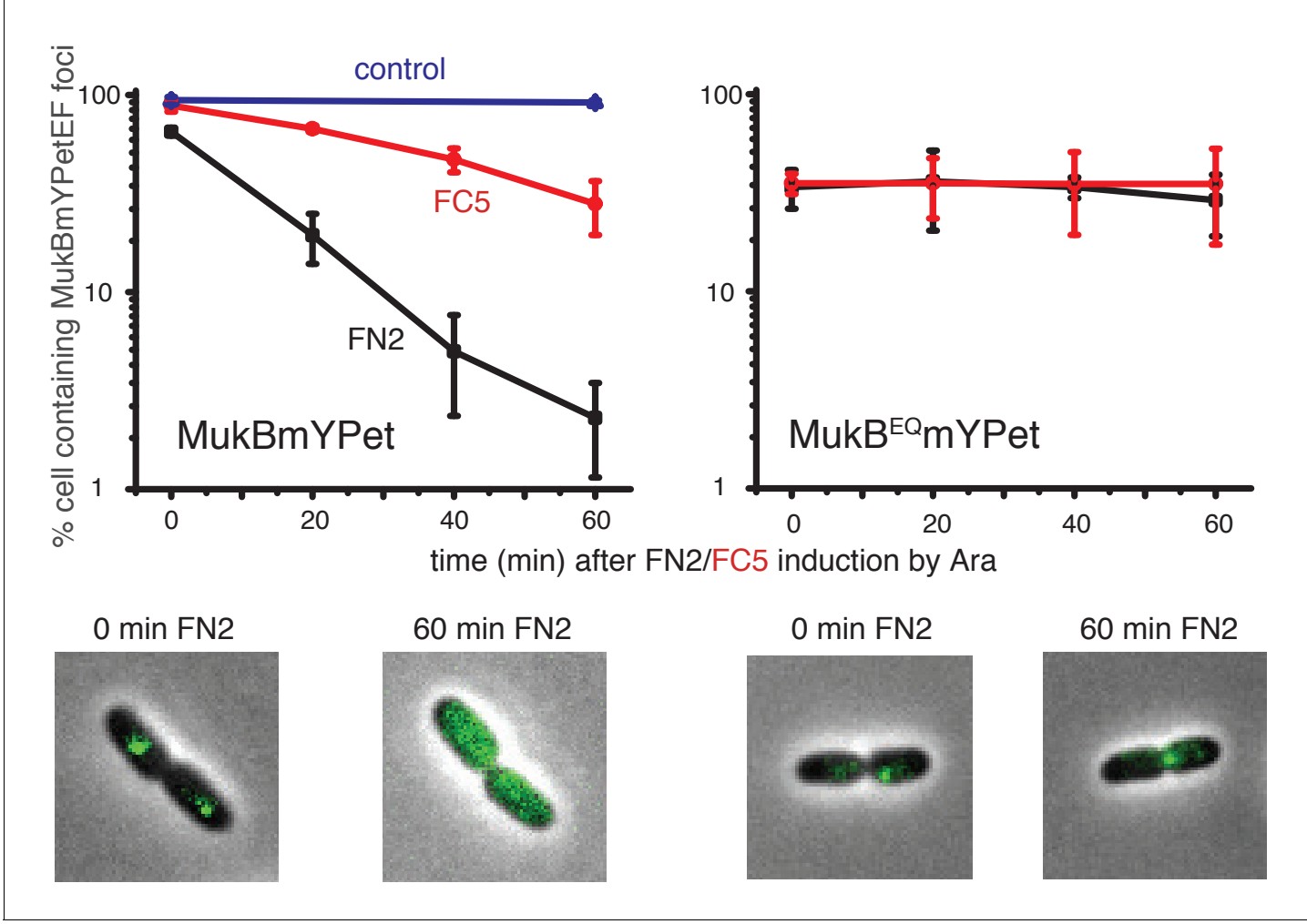

**Figure 8.** Overexpression of MukF N-terminal and C-terminal domains fragments leads to an ATP hydrolysis cycle-dependent release of MukBEF complexes from DNA in vivo. MukF FN2 and FC5 fragments were overexpressed from *para* promoter in pBAD24 by addition of arabinose. MukBmYPetEF and MukB$^{EQ}$mYPetEF complexes were visualised in the absence of arabinose and at every 20 min after induction. More than 500 cells were analysed for each condition. Experiment was repeated three times; error bars show standard deviation of 3 repeats. Bottom panel; images of FN2 overexpressing cells taken at time 0 and 60 min in MukBmYPetEF and MukB$^{EQ}$YPetEF strains (SN182 and SN311, *Nolivos et al., 2016*).
DOI: https://doi.org/10.7554/eLife.31522.031

The following source data and figure supplements are available for figure 8:

**Source data 1.** Overexpression of MukF N-terminal and C-terminal domains fragments leads to an ATP hydrolysis cycle-dependent release of MukBEF complexes from DNAin vivo.
DOI: https://doi.org/10.7554/eLife.31522.034

**Figure supplement 1.** MukBEF foci in cells carrying either *mukB* wt or *mukB$^{EQ}$mYPet* chromosomal genes, before and after 1 hr arabinose induced overexpression of MukF N- and C-terminal domain fragments, FN2 and FC5.
DOI: https://doi.org/10.7554/eLife.31522.032

**Figure supplement 2.** Expression levels of FN2 and FC5 after arabinose-induced induction.
DOI: https://doi.org/10.7554/eLife.31522.033

neck, or cap, respectively, the reminder of MukF remains associated with MukB through its other interactions, thereby giving a high re-binding rate.

FN2 over-expression led to a ~4 fold more efficient displacement of labelled MukBEF complexes from DNA, than over-expression of FC5. These observations of ATP hydrolysis-dependent release are consistent with the interface between the MukF N-terminal domain and the neck disengaging more frequently than the interface between MukF C-terminal domain and the cap during the activity

cycles of MukBEF. Alternatively, disruption of the MukF-neck interaction could lead to more imminent release of DNA from the complex. We note that at least one MukF C-terminal domain-MukB cap interface in a dimeric MukBEF complex has to be broken in each activity cycle to allow the formation of an asymmetric heads-engaged dimeric complex in which only one MukB monomer can bind a MukF C-terminus because of steric occlusion (*Shin et al., 2009*; *Woo et al., 2009*). Rebinding of a second MukF C-terminal domain to a MukB cap that becomes available after ATP hydrolysis might be necessary to initiate the next cycle of MukBEF activity. Interfaces between the kleisin N-terminus and the SMC neck in yeast, *drosophila* and human cohesin complexes have been proposed previously to function as DNA exit gates dependent on SMC ATP hydrolysis (*Beckouët et al., 2016*; *Buheitel and Stemmann, 2013*; *Chan et al., 2012*; *Eichinger et al., 2013*; *Huis in't Veld et al., 2014*).

## Discussion

The work here has revealed two important and related new insights into the action of the *E. coli* SMC complex, MukBEF. First, the demonstration that the MukF N-terminal domain interacts functionally with the MukB neck, with impairment of this interaction leading to MukBEF complex release from chromosomes and a Muk⁻ phenotype. Second, the observation that the MukF C- and N-terminal domains activate MukB ATPase independently and additively, with each domain contributing ~50% of the maximal activity in steady state assays. We propose that both of these properties relate to the formation of an asymmetric complex between a MukB dimer and MukEF after ATP binding and consequent MukB head engagement.

A crystal structure of a complex between a heads-engaged *Haemophilus ducreyi* MukB dimer bound by MukFE revealed this asymmetry (*Figure 1B*; pdb 3EUK; *Woo et al., 2009*; *Figure 9A*). In the asymmetric structure, only one MukB head was bound by a MukF C-terminal domain, with the adjacent MukF middle region binding to the second MukB monomer of the MukB dimer, thereby sterically occluding the binding of a second MukF C-terminal domain. This asymmetry induced by MukB head engagement was also observed in solution (*Woo et al., 2009*). Because the MukF N-terminal domain was absent in the variant used, the interaction uncovered here between the MukF N-terminal domain and the MukB neck was not evident. This view of an asymmetric MukBEF complex when heads are engaged is supported by biochemical and in vivo studies (*Shin et al., 2009*; *Badrinarayanan et al., 2012a*). Furthermore, such an asymmetry directed by interaction of the C- and N-terminal domains of kleisin with the head and neck of SMC dimers appears to be functionally conserved, regardless of whether they form SMC homodimers or heterodimers (*Bürmann et al., 2013*; *Gligoris, 2014*; *Huis in't Veld et al. 2014*).

However, unlike other kleisins, which are apparently monomeric, MukF is a stable dimer and its dimerisation domain is adjacent to the 4-helix bundle, to which helices 8 and 9 belong. Nevertheless, binding of these helices to the MukB neck did not interfere with MukF dimerisation, a result consistent with our previous analysis inferring the existence of MukBEF dimers of dimers in vivo (*Badrinarayanan et al., 2012a*; *Figure 9A*; top panel; right). Note that an asymmetric heads-engaged MukBEF dimer can potentially form a symmetric dimer of dimers as a consequence of MukF dimerisation.

We propose that in an engaged-heads dimeric MukBEF complex that the MukF C- and N-terminal domains contained within a single MukF molecule bind separate MukB monomers in the dimer (*Figure 9A*; top panel-trans), as demonstrated for other SMC complexes. Nevertheless, we cannot exclude the possibility that the asymmetric complex has the MukF C- and N-terminal domains bound to only one of the two MukB molecules, since we have shown that a single monomer of MukB_{HN} can interact with separated MukF N- and C-terminal domains. If this were the case, an additional asymmetric interaction of the MukF middle region with the MukB monomer that is not bound by the MukF C- and N-terminal domains would be present (*Figure 9A*; top panel; cis). Examination of the heads-engaged asymmetric MukBEF crystal structure, in which the MukF N-terminal is absent, does not allow us to distinguish between these possibilities.

Interaction between the MukF N- terminal domain and MukB neck is not only essential for MukBEF function in vivo, but also appears to be broken and reformed during cycles of ATP binding and hydrolysis. Additionally, interaction between the MukF C-terminal domain and the MukB cap may also break and reform during ATP binding and hydrolysis cycles. The observation that impairment of

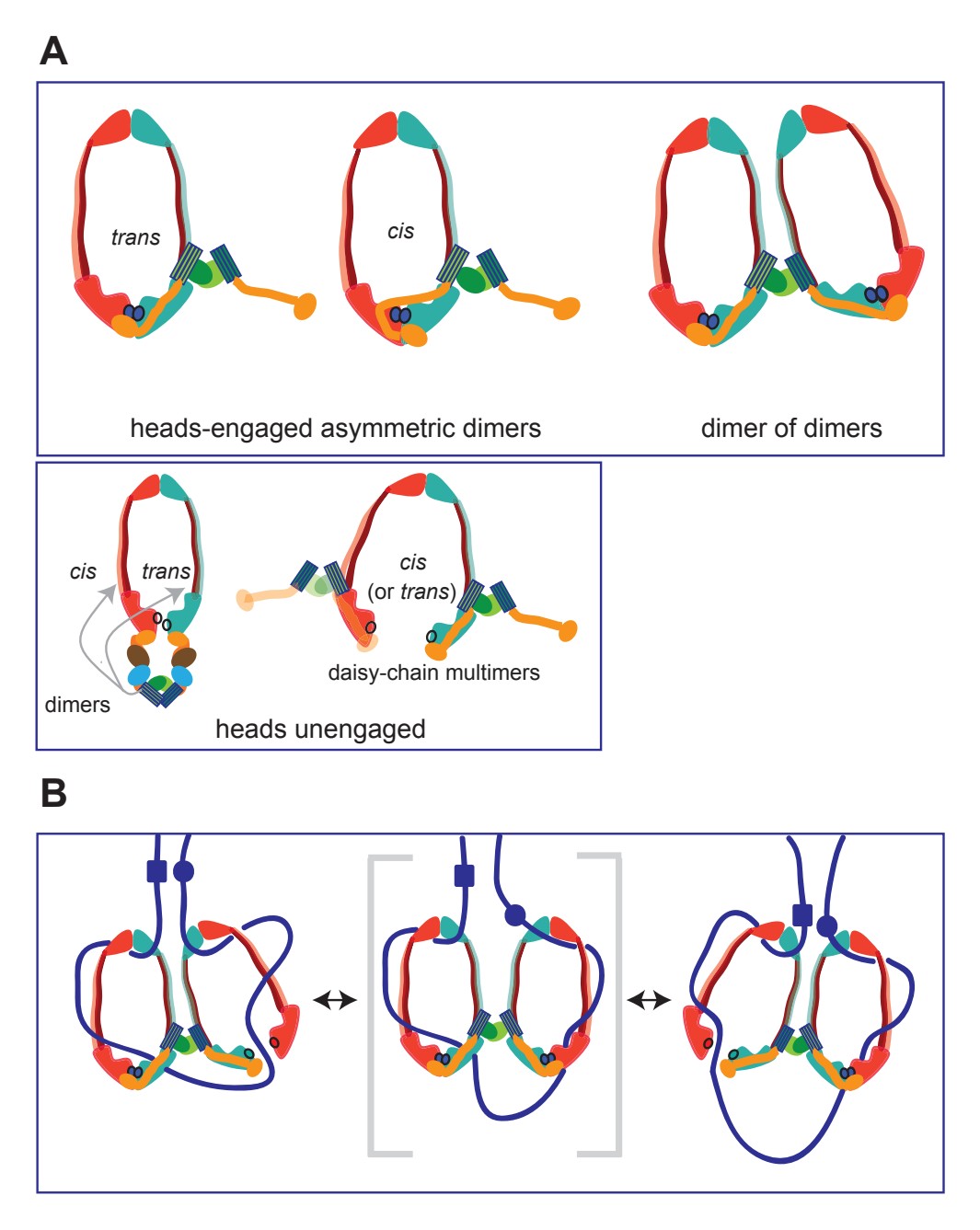

**Figure 9.** Summary of MukBF interactions and a model for DNA transport. (**A**) Top panel; Schematics of possible configurations of MukB and MukF interactions in ATP-bound head-engaged asymmetric complexes of MukBEF. We favour the *trans*-configuration, identical to those of other SMC complexes. Bottom panel; possible unengaged head complexes. The cartoon on the left is a repeat of that in *Figure 1*, but additionally indicating that the MukF 4-helix bundles may interact with the same MukB molecule as its C-terminal domain (*cis*), or the partner MukB molecule (*trans*). On the right, is a cartoon indicating how daisy-chained multimers could form; these have not been detected in the studies here. (**B**) Rock- (or rope-) climber model for DNA transport and loop extrusion by MukBEF, modified from (*Badrinarayanan et al., 2012a*). For clarity only MukBF is shown. The paths of DNA (blue lines; not to scale) are hypothetical, although DNA interactions with the MukB head and hinge have been demonstrated (*Kumar et al., 2017*; *Woo et al., 2009*). The state shown in the middle panel (parentheses) could rarely exist if ATP binding and hydrolysis were to be coordinated between the two MukBEF dimers. For presentational simplicity, we have shown MukBEF ring opening through head disengagement, by release and transfer of the MukF C-terminal domain to the '*cis*-configuration', although results here and elsewhere provide stronger support for ring opening through release of the MukF N-terminal domain from the MukB neck. In reality, the patterns of DNA binding and release, and the conformational changes in the complexes are likely to be more complicated, with both an upper and a lower chamber in each dimeric complex (for example, see *Diebold-Durand et al., 2017*; *Uhlmann, 2016*).

*Figure 9 continued on next page*

*Figure 9 continued*

DOI: https://doi.org/10.7554/eLife.31522.035

the normal MukF-MukB interactions leads to ATP hydrolysis-dependent loss of MukBEF clusters from chromosomes indicates that by opening of at least one MukB-MukF interface, DNA can be released from the 'bottom ring chamber' formed by a kleisin bridging a MukB head and the MukB neck of a partner molecule. This result provides further support for a mechanism in which ATP hydrolysis is required to release MukBEF and other SMC complexes from chromosomes (*Murayama and Uhlmann, 2015*; *Nolivos et al., 2016*). Equivalent interfaces between the kleisin and SMC3 neck in the yeast, *drosophila* and human cohesin complexes have also been proposed to act as DNA exit gates and it has been proposed that this interaction, which is not required for loading onto chromosomes, turns-over in response to ATP binding and hydrolysis (*Beckouët et al., 2016*; *Buheitel and Stemmann, 2013*; *Chan et al., 2012*; *Eichinger et al., 2013*; *Elbatsh et al., 2017*).

Although, a DNA exit gate formed by the SMC coiled-coil neck-kleisin interaction appears to be conserved, we think it possible that other interfaces could additionally be used for DNA release. For example, the hinge dimerisation interface, which has been proposed to be a DNA entrance gate (*Buheitel and Stemmann, 2013*; *Gruber et al., 2006*), might additionally function as an exit gate under some conditions (*Murayama and Uhlmann, 2013*; *Uhlmann, 2016*). Because there are two potential proteinaceous chambers in SMC complexes, the upper one formed by a heads-engaged SMC complex and the lower one by the kleisin bound to the SMC (*Diebold-Durand et al., 2017*; *Uhlmann, 2016*), each of these chambers could have exit (and entrance) gates for DNA segments entrapped within each of them. In MukBEF, interaction of MatP-*matS* with the MukB hinge has been proposed to promote ATP hydrolysis-dependent release of MukBEF clusters from the *ter* region of the chromosome, suggestive of release through the dimerisation hinge (*Nolivos et al., 2016*). Similarly, MukB-dependent stimulation of catalysis by TopoIV could arise as a consequence of DNA exiting the MukB hinge and being presented to the TopoIV entrance gate, which is in proximity to the MukB hinge (*Vos et al., 2013*; *Zawadzki et al., 2015*).

We propose that the observed independent and additive regulation of MukB ATPase by the MukF C- and N-terminal domains, may reflect asymmetry in the two ATPases resulting from asymmetric heads-engaged MukBEF dimer complexes. One of these could be activated by the MukF N-terminal domain and the other by the C-terminal domain. Nevertheless, examination of the heads-engaged MukBEF crystal structure did not reveal any differences in the two ATPase active sites. How the C- and N-terminal MukF domains activate MukB ATPase remain unclear; but their independent and additive action through interaction with different MukB targets, most likely on separate MukB molecules, is in our opinion, most consistent with them activating separate ATPase sites in a MukB dimer. Whether these activations occur at the stage of ATP binding, head engagement, the actual catalytic step, or several of these, remains to be determined.

Asymmetric ATPase mechanisms have been demonstrated for ABC transporters, which share the overall organisation of their ATPase heads with SMCs (*ter Beek et al., 2014*; *Procko et al., 2009*; *Zhou et al., 2016*). Similarly, eukaryotic heterodimeric SMC complexes have active site region differences; for example, comparison of SMC1 sequences with those of SMC3 show protein family-specific differences, with their ATPases being differentially regulated, and in at least some cases, with independent functions (*Beckouët et al., 2016*; *Çamdere et al., 2015*; *Elbatsh et al., 2017, 2016*).

We have inferred previously from in vivo experiments that ATP hydrolysis by MukBEF is required for both loading and unloading onto DNA (*Nolivos et al., 2016*). Therefore, one plausible explanation of our data is that the MukB neck-kleisin interaction acts in DNA unloading, leaving the MukB head-kleisin interaction to function in loading onto chromosomes. Similarly, a functional asymmetry in yeast cohesion ATPase active sites has been proposed, with the one equivalent to the kleisin-neck interaction uncovered here being required for release from chromosomes, while both ATPases were implicated in loading onto chromosomes (*Çamdere et al., 2015*; *Elbatsh et al., 2017*). Our own data do not address whether the interaction with the MukB neck is also required for loading.

The significance of how and why MukE inhibits the MukB ATPase activated by either the MukF C-terminal, or the N-terminal domains, remains unclear. It could arise simply from the fact that MukE binding stabilises a particular MukBF conformation, thereby leading to less turnover during the

steady state multiple turnover ATPase assays. Alternatively, or additionally, this could reflect MukE playing a regulatory role during transitions between various stages of MukBEF activity cycle. Other in vitro and in vivo studies have postulated a regulatory role of MukE (*Gloyd et al., 2011*; *She et al., 2013*), although details of how this regulation is mediated have been unclear. Nevertheless, depletion of MukE in vivo mimics the ATP hydrolysis-impaired phenotype of a MukB$^{EQ}$ mutant, which loads slowly onto DNA in the *ter* region, but is unable to undergo the multiple cycles of ATP binding and hydrolysis required to target to *ori* (*Badrinarayanan et al., 2012b*; *Nolivos et al., 2016*).

The ability of DNA to relieve MukE inhibition of MukB ATPase could result from MukE and DNA competing for binding to the MukB head. This is consistent with in vitro studies, which showed competition between MukEF and DNA for MukB binding and that MukEF inhibited MukB-mediated DNA condensation (*Cui et al., 2008*; *Petrushenko et al., 2006b*). Furthermore, a patch of positively charged amino acid residues on the surface of MukB head, close to the base of the neck, was shown to be important for interaction with DNA (*Figure 7C*; *Woo et al., 2009*). Projection of B-form DNA onto this patch highlights the potential competition of DNA- and MukE-binding to a MukBF complex, which may reflect alternative states during the MukBEF-DNA activity cycle.

A range of structures, alongside extensive biochemical and functional analyses, leads to the conclusion that all SMC complexes, including MukBEF, share distinctive architectures and similarities in their likely molecular basic mechanisms of action on chromosomes. Central to the SMC complex mechanism is the ability to bind and hydrolyse ATP in a modulated fashion, which directs stable loading onto chromosomes, regulated release from chromosomes and autonomous rapid transport with respect to DNA, which is likely to depend on such loading and release (*Diebold-Durand et al., 2017*; *Terkawa et al., 2017*; *Wang et al., 2017*). Any such transport must require at least two specific DNA-SMC complex attachment points on different conformational states of the complex, with coordinated transitions as transport proceeds. Our finding that MukF dimerisation is maintained during its interaction with the MukB neck, not only validates our demonstration of dimers of MukBEF dimers in active MukBEF clusters in vivo, but provides support for our previously proposed 'rock (or rope) climber' model for the transport of MukBEF dimer of dimers with respect to DNA (*Figure 9B*; *Badrinarayanan et al., 2012a*). This model assumes that dimers of MukBEF dimers are a minimal functional unit, in which coordinated capture and processing of DNA segments by each MukBEF dimer, similar to the action of a climber reaching out to 'grab' a rock/rope alternatively with each arm. The staggered cycles of ATP binding and hydrolysis, DNA trapping and release and associated conformational changes could effectively coordinate the activity within the partner dimers within MukBEF dimers. For SMC complexes that do not obviously form dimers of dimers, the type of loop-capture and fusion model proposed by *Diebold-Durand et al., 2017* has merits. In this, DNA loops captured in the upper SMC chamber are transferred to the lower chamber, where they fuse with a pre-existing loop, thereby meeting the basic requirements for ATP hydrolysis-driven transport.

## Materials and methods

**Key resources table**

| Reagent type (species) or resource | Designation | Source or reference | Identifiers |
|---|---|---|---|
| strain, strain background (*E.coli* K12 AB1157) | mukBmYPetEF | SN 182 Nolivos et al.; DOI: 10.1038/ncomms10466 | |
| strain, strain background (*E.coli* K12 AB1157) | mukB EQ mYPetEF | SN 182 Nolivos et al.; DOI: 10.1038/ncomms10466 | |
| Strain, for protein expression (*E.coli* C3013I) | | NEB | |
| strain, for MukB protein expression (*E.coli* C3013I) C3013I - mukB 3xflag tag | mukB 3xFLAG C3031 | FLOI - this work; derivative of SN 54; Nolivos et al; DOI: 10.1038/ncomms10466 | |
| strain, strain background (*E.coli* K12 AB1157) | RRL 149; Δ mukB | Nolivos et al.; DOI: 10.1038/ncomms10466 | |
| strain, strain background (*E.coli* K12 AB1157) | Ab 233; Δ mukF mukBGFP | Nolivos et al.; DOI: 10.1038/ncomms10466 | |

*Continued on next page*

*Continued*

| Reagent type (species) or resource | Designation | Source or reference | Identifiers |
|---|---|---|---|
| antibody | anti MukF- (mouse polyclonal) | gift from Kenneth Marians, Memorial Sloan Kettering Center, New York | |
| antibody | Anti-Mouse IgG (whole molecule)–Peroxidase antibody produced in goat | SIGMA | RRID: AB_258167 |
| commercial assay or kit | ENZCheck Phosphate Assay | Life Technologies | |
| software, algorithm | ASTRA 6 | Wyatt Technologies | |
| software algorithm | MicrobTracker-Matlab | *Sliusarenko et al. (2011)* | RRID:SCR_001622 |
| software, algorithm | Pymol | https://pymol.org/2/ | RRID:SCR_000305 |
| software, algorithm | Methamorph - Ni elements | Nikon | |
| software, algorithm | Modeller | | RRID:SCR_008395 |
| software algorithm | MARS data analysis | BMG Lagtech | |

## Protein purification

MukB, MukB$_H$, MukB$_{HN}$, MukE, were 6xHis-tagged at the C-terminus (pET21), while MukF and its C- and N-terminal truncations were 6xHis-tagged at the N-terminus (pET28). MukB variants were expressed in strain FL01, which is mukB 3xFLAG C3013I (NEB). MukF variants and MukE 6xHis-tagged at the C-terminus were expressed from pET21 in C3013I cells (NEB). 2L cultures were grown in LB with appropriate antibiotics at 37°C to $A_{600}$ ~0.6 and induced by adding IPTG at final concentration of 0.4 mM. After 2 hr at 30°C, cells were harvested by centrifugation, re-suspended in 30 ml lysis buffer (50 mM HEPES pH 7.5, 300 mM NaCl, 5%glycerol, 10 mM imidazole) supplemented with 1 tablet of protease inhibitor (PI), and homogenised. Cell debris was removed by centrifugation and clear cell lysates were mixed with 5 ml equilibrated TALON Superflow resin, poured into a column, then washed with 10 X volume of washing buffer (50 mM HEPES pH 7.5, 300 mM NaCl, 5% glycerol, 25 mM imidazole, PI). Bound proteins were eluted in elution buffer (50 mM HEPES pH 7.5, 300 mM NaCl, 5% glycerol, 250 mM imidazole). The fractions from TALON were diluted to 100 mM NaCl buffer and injected to HiTrapTM Heparin HP column (GE Healthcare) pre-equilibrated with Buffer A (50 mM HEPES pH 7.5, 100 mM NaCl, 10% glycerol, 1 mM EDTA, 1 mM DTT), then the column was washed at 1 ml/min flow rate until constant $A_{280}$. Purified fractions were eluted with a gradient 100–1000 mM NaCl.

For MukE and MukF purifications, fractions from Talon were diluted and injected onto a HiTrap DEAE FF column (GE healthcare) pre-equilibrated in Buffer A. Purified fractions were eluted with a gradient 100–1000 mM NaCl.

Protein concentration was estimated by UV absorption at 280 nm on Nanodrop spectrophotometer, and protein purity and identity confirmed by electrospray ionisation mass-spectrometry and SDS PAGE. Proteins were aliquoted and stored at −20°C in a buffer containing 10% glycerol.

## ATP hydrolysis assays

ATP hydrolysis was analysed in steady state reactions using an ENZCheck Phosphate Assay Kit (Life Technologies). 150 μL samples containing standard reaction buffer supplemented with 2 mM of ATP were assayed in a BMG Labtech PherAstar FS plate reader at 25°C. The data were analysed using MARS data analysis software. Quantitation of phosphate release was determined using the extinction coefficient of 11,200 $M^{-1}cm^{-1}$ for the phosphate-dependent reaction at $A_{360}$ nm at pH 7.0.

## Size exclusion chromatography and Multi-Angle light scattering (SEC-MALS)

Purified proteins were fractionated on a Superose 6 10/300 GL or a Superose 12 10/300 column equilibrated with 50 mM HEPES, pH 7.5 buffer containing 100 mM NaCl, 1 mMDTT, 1 mM EDTA, at flow rate of 0.5 ml/min. 500 μl samples containing analysed proteins were injected on the column and run at a flow rate of 0.5 ml/min. SEC-MALS analysis was performed at 20°C using a Shimadzu (Kyoto, Japan) chromatography system, connected in-line to a Heleos8+ multi angle light scattering detector and an Optilab T-rEX refractive index (RI) detector (Wyatt Technologies, Goleta, CA).

Protein samples in 50 mM HEPES pH 7.5, 100 mM NaCl, 1 mM DTT, 1 mM EDTA, 10% glycerol, were injected in this system, and the resulting MALS, RI and UV traces processed in ASTRA 6 (Wyatt Technologies).

## Pull-down assays

MukF FLAG-tagged fragments were expressed from pET DUET plasmids in C3013I cells (NEB).1L cultures were grown in LB with carbenicilin (100 µg/ml) at 37°C to $A_{600}$ ~0.6 and induced by adding IPTG to a final concentration 0.4 mM. After 2 hr at 30°C, cells were harvested by centrifugation, re-suspended in 30 ml lysis buffer (50 mM HEPES pH 7.5, 300 mM NaCl, 5%glycerol, 10 mM imidazole) supplemented with 1 tablet of protease inhibitor (PI), and homogenised. Cell debris was removed by centrifugation and clear cell lysates were mixed with 150 µl Anti-FLAG M2 Affinity gel (Sigma Aldrich), incubated for 1 hr at 4°C. The resin was then washed three times with the same buffer containing 250 mM NaCl, resuspended in 1 ml of buffer I (50 mM HEPES pH 7.5, 100 mM NaCl), and purified MukB, $MukB_H$ or $MukB_{HN}$ were added. After 45 min incubation (4°C) the resin was washed three times, re-suspended in 200 µl of protein loading buffer (NEB) and analysed on 4–20% gradient SDS PAGE.

## Rationale for targeted mutagenesis

Our design of amino acid substitutions in the coiled-coil of MukB neck and MukF helix 9 was informed by the arrangement and interactions at the interface between Scc1 kleisin helices that interact with the coiled-coil of cohesin (Gligoris et al., 2014). Both SMC coiled-coil helices, one protruding from N-terminal and the other from the C-terminal subdomain interact with Scc1. We targeted the MukB C-terminal helix for mutagenesis because we could make better predictions for the orientation of this helix in the MukB neck. Three sets of double mutations were constructed; they mapped to the same side of the helix, but with each set on a slightly different face.

The rationale for mutagenesis in MukF helix 9 was to mutate solvent exposed residues that were not predicted to interact with other MukF helices, or be obstructed by MukE binding. Three sets of triple mutations along the helix were constructed. All amino acid residues chosen for mutagenesis were either invariant or very highly conserved among bacterial species. Point mutations in plasmid-encoded genes were made using Q5 site-directed mutagenesis Kit (NEB). Primers were designed with NEBase Changer. 10 ng of the template was taken to the reaction. Plasmids were isolated and mutations confirmed by sequencing.

## Complementation assays

The ability of leaky plasmid-encoded MukF or MukB expression from pET21, in the absence of IPTG, to complement the temperature-sensitive growth defect of ΔmukF (AB 233) or ΔmukB (RRL149) cells, respectively, at 37°C in LB was assayed. Cells were transformed with pET21 carrying MukF or MukB, or their variants, and allowed to recover for 8 hr post transformation at permissive temperature then plated in duplicates on LB plates containing carbenicillin (100 µg/ml). One plate was incubated at non-permissive (37°C) and the other one at permissive (20°C) temperature. Colonies from plates incubated at permissive temperature were streaked in duplicate and grown at permissive and non-permissive temperature along with positive and negative controls.

## Analysis of MukBEF function in vivo

Strains were streaked onto LB plates with appropriate antibiotics. Single colonies were inoculated into M9 glycerol (0.2%) and grown overnight at 37°C to $A_{600}$0.4–0.6, then diluted into fresh M9 and grown to $A_{600}$ 0.1. Cells were spun and immobilised on agarose pads between two glass coverslips (1.5 thickness). 1% agarose pads were prepared by mixing low-fluorescence 2% agarose (Bio-Rad) in $dH_2O$ 1:1 with 2x growth medium. For analysis of MukBEF fluorescent clusters (foci), strains carrying either functional MukBmYPet (SN182), or the ATP hydrolysis-impaired mutant $MukB^{EQ}$mYPet (SN311, Nolivos et al., 2016) were used. Wide-field fluorescence microscopy used an Eclipse TE2000-U microscope (Nikon), equipped with an 100x/NA1.4 oil PlanApo objective and a Cool-Snap $HQ^2$ CCD, and using Metamorph software for image acquisition. Over-expression of FN2 and FC2 was from pBAD24 plasmids containing the appropriate arabinose-inducible MukF derivative. Strains were transformed with given plasmid and grown in M9 glycerol medium supplemented with 0.2%

glucose to limit leaky expression from the arabinose promoter. Once cultures reached $A_{600}$ ~0.1, cells were centrifuged and re-suspended in M9 glycerol medium supplemented with 0.2% L-Arabinose and grown at 37°C. Every 20 min, cells from 1 ml of culture were taken, centrifuged, placed on agarose pad and imaged. As a control, strain carrying empty pBAD24 vector was analysed. Cells were segmented from brightfield images using MicrobeTracker (*Sliusarenko et al., 2011*). MukB foci were detected using 'spotfinderM', available as part of the MicrobeTracker Suite.

## Fluorescence correlation spectroscopy (FCS)

FCS was carried out on a ConfoCor 2 system (Carl Zeiss). The 633 nm line of a HeNe laser was directed via a 488/561/633 dichroic mirror and focused with a Zeiss C-Apochromat 40 Å~NA 1.2 water immersion objective to excite experimental samples containing Cy5. Fluorescence emission was collected using a 655 nm long pass filter and recorded by an avalanche photodiode. The pinhole diameter was adjusted to 83 μm (one Airy unit), and the pinhole position was optimised with use of the automatic pinhole adjustment for Cy5. All FCS experiments were carried out in Lab-Tek (Nagle Nunc International) eight-well chambered borosilicate glass plates at 22 ± 1°C. In the assay, diffusion of Cy5-labelled FN2 and FN10 fragments at fixed concentrations (~10 nM) was measured in samples carrying MukB at a range of concentrations up to 160 μM. Since MukB is much larger than any of the fragments used, up to a 3-fold increase in diffusion time was observed.

The intensity of fluorescence signal was measured and the autocorrelation function G(t) was determined for diffusing fluorescently labelled species present in the sample. If two species with different diffusional properties are present, the autocorrelation function G(t) can be described as a two-component model that allows analysis of the abundance of each species:

$$G(\tau) = \left[ 1 - T + T \exp\left(\frac{-\tau}{\tau T}\right) \right] N^{-1} = \left[ \frac{1-Y}{\left(1+\frac{\tau}{\tau_{substrate}}\right)\sqrt{1+\frac{r_0^2}{z_0^2}\frac{\tau}{\tau_{substrate}}}} + \frac{Y}{\left(1+\frac{\tau}{\tau_{product}}\right)\sqrt{1+\frac{r_0^2}{z_0^2}\frac{\tau}{\tau_{product}}}} \right]$$

where T is the average fraction of dye molecules in the triplet state with the relaxation time τT, N is the average number of fluorescent molecules in the volume observed, Y is the relative fraction of fragment bound to MukB, τ substrate and τ product are the diffusion time constants of free protein (labelled fragment as indicated for individual experiment and fragment bound to MukB), respectively, and r0 and z0 are the lateral and axial dimensions, respectively, of the observation volume. All calculations, including the evaluation of the autocorrelation curves, which was carried out with a Marquardt nonlinear least-square fitting procedure, were performed using the ConfoCor 2 instrument software. To obtain the % of bound and unbound fragments, the diffusion times for fluorescently labelled fragment were measured and fixed during data analysis. The diffusion time for the complex of a given fragment and MukB was estimated based on measured diffusion time for labelled MukB. No change in diffusion time for labelled MukB was observed when unlabelled fragment was added; therefore, the measured diffusion time for MukB was used as a fixed value during data analysis.

## Fluorescence polarization anisotropy (FPA)

Experiments were done on a BMG LABTECH PHERAstar FS next-generation microplate reader with an FP 590–50 675–50 optic module. Samples were measured in Corning black 96 well flat bottom half volume plates at 25°C. All sample volumes were 100 μL. Cy5 labelled FN3 and FC2 were used at 5 nM and 9 nM respectively. The concentration of MukB was varied from 0.1 nM to 1 μM. Samples were equilibrated for 40 min before measurement. Experiments were repeated thrice and standard deviations are reported. Data were plotted and analysed using Sigmaplot, where $K_d$ and total receptor concentration were solved simultaneously. Binding reached saturation above 160 nM MukB. Binding of FN3 or FC2 with 1 μM MukB was used as a 100% bound reading. The fraction of FN3 or FC2 bound was determined using the equation:

$$\left[ 1 - \left( \frac{Max\ value - Current\ value}{Max\ value - Min\ value} \right) \right] * 100\%$$

Data were plotted and the values of $K_d$ and 'total receptor' concentration ($R_T$) were simultaneously determined using Sigmaplot by solving the quadratic for fraction bound (B) below,

$$B = \frac{(MukB_T + K_d + R_T) - \sqrt{((-MukB_T - K_d - R_T)^2 - 4MukB_T R_T)}}{2}$$

## Western blot analysis

MukB$^+$ (SN182) and MukB$^{EQ}$ mutant (SN311) cells were transformed with pBAD, pBAD-FN2 (pKZ111) and pBAD-FC5 (pZ103) plasmids. Cells were grown at 22°C, to A$_{600}$0.4–0.6, induced with 0.2% L-Arabinose for 3 hr. Cultures were spun down and cell pellets were resuspended in gel loading buffer and proteins were separated by a 4–20% gradient SDS PAGE followed by Western blots with mouse anti-MukF antibody as primary and goat anti-mouse as secondary antibody.

## Quantitative mass spectroscopy

In-solution trypsin digestion. Bacterial lysates were prepared from 50 ml cultures grown in M9 minimum media to A$_{600}$ ~0.2, with expression from pBAD- induced with 0.2% arabinose for 1 hr. Cells were centrifuged and the pellet was resuspended in 200 µL of 0.1% SDS in PBS, sonicated and incubated for 5 min at 100°C. After centrifugation, supernatant was collected and protein concentration was as assessed using the BCA (Thermo Fisher Scientific, USA) method. Then, 10 µg of protein extract was digested with 0.2 µg of sequencing-grade trypsin (Promega, Mannheim, Germany) overnight at 37°C Proteins were reduced with DTT and alkylated using iodoacetamide. Each sample was prepared for digestion in duplicate.

NanoLC-MS/MS Analysis. For each run, 1.5 µg of the digested protein samples was injected onto an RP C18 precolumn (Thermo Fisher Scientific, Waltham, MA, USA) connected to a 75 µm i.d. ×25 cm RP C18 Acclaim PepMap column with a particle size of 2 µm and a pore size of 100 Å, using a Dionex UltiMate 3000 RSLCnano System (Thermo Fisher Scientific). Every sample was injected in duplicate at random. Before analysis, the system was calibrated using Pierce LTQ ESI Positive Ion Calibration Solution (Thermo Fisher Scientific). The following LC buffers were used: buffer A (0.1% (v/v) formic acid in Milli-Q water) and buffer B (0.1% formic acid in 90% acetonitrile). The peptides were eluted from the column with a constant flow rate of 300 nL.min$^{-1}$ with a linear gradient of buffer B from 5% to 65% for 120 min. At 100 min, the gradient was increased to 90% B and was held there for 10 min. Between 110 and 120 min, the gradient returned to 5% to re-equilibrate the column for the next injection. The peptides eluted from the column were analysed in the data-dependent MS/MS mode on a Q-Exactive Orbitrap mass spectrometer (Thermo Fisher Scientific). The instrument settings were as follows: the resolution was set to 70,000 for MS scans, and 17,500 for the MS/MS scans to increase the acquisition rate. The MS scan range was from 300 to 2000 m/z. The MS AGC target was set to 1 × 106 counts, whereas the MS/MS AGC target was set to 5 × 104. Dynamic exclusion was set with a duration of 20 s. The isolation window was set to 2 m/z.

Analysis of proteomic data. After each LC-MS/MS run, the raw files were analysed by Proteome Discoverer, version 1.4.14 (Thermo Fisher Scientific). The identification of proteins was performed using the MASCOT engine against the UniProt after adding to database sequences of recombinant proteins FN2 and FC5. Analyses were completed using the following parameters: a tolerance level of 10 ppm for MS and 0.05 Da for MS/MS and with 1% FDR. Trypsin was used as the digesting enzyme, and one missed cleavage was allowed. Estimation of protein abundance was based on the emPAI parameter (*Ishihama et al., 2005*).

## Acknowledgements

The work was supported by a Wellcome Trust Senior Investigator Award to DJS (099204/Z/12Z) and by the Leverhulme Trust (RP2013-K-017). PZ was also supported by a National Science Centre, Poland grant: 2015/19/P/NZ1/03859 and by a Foundation for Polish Science grant: First TEAM/ 2016-1/9. We thank all past and present DJS laboratory members for illuminating discussions.

## Additional information

### Competing interests

David J Sherratt: Reviewing editor, *eLife*. The other authors declare that no competing interests exist.

### Funding

| Funder | Grant reference number | Author |
|---|---|---|
| Wellcome Trust | Senior Investigator Award (099204/Z/12Z) | David J Sherratt |
| Leverhulme Trust | RP2013-K-017 | David J Sherratt |
| Narodowe Centrum Nauki | 2015/19/P/NZ1/03859 | Pawel Zawadzki |
| Fundacja na rzecz Nauki Pols-kiej | First TEAM/2016-1/9 | Pawel Zawadzki |

The funders had no role in study design, data collection and interpretation, or the decision to submit the work for publication.

### Author contributions

Katarzyna Zawadzka, Data curation, Formal analysis, Investigation, Methodology, Carried out most of the experimental work, including construction of protein variants, pull-down assays, SEC-MALS, ATPase, and FCS assays, Contributed to data interpretation and presentation; Pawel Zawadzki, Conceptualization, Formal analysis, Investigation, Visualization, Performed in vivo analyses and contributed to FCS analysis and proteomics, Contributed to data interpretation and presentation; Rachel Baker, Formal analysis, Investigation, Carried out the SEC analyses, ATPase and in vivo complementation assays, Contributed to data interpretation and presentation; Karthik V Rajasekar, Data curation, Formal analysis, Carried out FPA assays, Western blot analysis and structure modelling, Contributed to data interpretation and presentation; Florence Wagner, Assisted in FLOI strain construction and in protein purification during early stages of the project; David J Sherratt, Conceptualization, Supervision, Funding acquisition, Writing—original draft, Project administration, Writing—review and editing; Lidia K Arciszewska, Conceptualization, Formal analysis, Supervision, Investigation, Methodology, Writing—original draft, Project administration, Writing—review and editing

### Author ORCIDs

Karthik V Rajasekar (iD) http://orcid.org/0000-0002-8146-6560
David J Sherratt (iD) https://orcid.org/0000-0002-2104-5430
Lidia K Arciszewska (iD) http://orcid.org/0000-0002-0252-4874

### Decision letter and Author response

Decision letter https://doi.org/10.7554/eLife.31522.038
Author response https://doi.org/10.7554/eLife.31522.039

## Additional files

### Supplementary files

• Transparent reporting form
DOI: https://doi.org/10.7554/eLife.31522.036

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
