## [Decision Letter]

Thank you for submitting your article "Asymmetric MukB ATPases are regulated independently by the N- and C-terminal domains of MukF kleisin" for consideration by *eLife*. Your article has been reviewed by three peer reviewers, and the evaluation has been overseen by a Reviewing Editor and Andrea Musacchio as the Senior Editor. The reviewers have opted to remain anonymous.

The reviewers have discussed the reviews with one another and the Reviewing Editor has drafted this decision to help you prepare a revised submission.

Summary:

The manuscript by Zawadzka et al. investigates the architecture of the *E. coli* MukBEF complex, its implications for the ATPase cycle and cellular functions. SMC complexes are genomic linkers that co-entrap DNA double helices using a large protein structure. Most SMC complexes form tripartite protein rings by the association of two SMC proteins with one kleisin subunit. In case of MukBEF, the exact architecture is less clear. MukF is an unusual member of the family of kleisin proteins. It forms homodimers via the interaction of its extreme N-terminal domain. In contrast, the kleisin subunits of other SMC complexes are known to interact with the head-proximal SMC coiled coil to form one of the three SMC ring interfaces.

All three reviewers agree that the manuscript provides solid and important new insights into MukBEF biochemistry and that it should appeal to a wide readership. However, the reviewers also raised a number of points, many of which can be addressed through textual changes or changes to the figures, as detailed below. Reviewer 2 found the manuscript very hard to read, and recommends that you work on your figures to make them more comprehensible. The reviewer also would like to see shorter figure legends and a substantially shortened Discussion section. There is also a general feeling that a cartoon drawing of the 'rock climber' model based on the authors' new findings would be desirable.

A strong objection, underlined by all three reviewers (and discussed again at points 7 and 8 of the "Required experimental additions" section), concerns the claim that MukB-F contains two independent ATPase domains. The reviewers believe that the data fail to demonstrate that the architecture of holo-MukBEF complexes is asymmetric. It is possible that a given MukF dimer binds symmetrically to either a single MukB dimer or to a dimer of MukB dimers. It is also unclear, whether FN or FC regulate the two MukB head domains differentially. The presented data is consistent with symmetric ATP hydrolysis by MukB. For instance, binding to FN or to FC might promote MukB head engagement rather than the ATP hydrolysis step. Thus, it seems premature to entertain different timing or different purpose for ATP hydrolysis by the one or the other MukB head domain. It is especially important that, among the other points, you adequately address this specific concern.

Required experimental additions:

1) Figure 2. In this semi-quantitative interaction assay it appears that interaction of the FN2, FN3 and FN4 deletion proteins with MukB is sequentially weakened, in this order. This suggests that the N-WHD and helixes 6 and 7 also make a contribution to MukB interaction. Is this the case? Clarification could come from the inclusion of FN4 in comparison with FN2 in the quantitative interaction assay (shown in Figure 3—figure supplement 3). Else, this should at least be discussed.

2) In addition, in this experiment, there also appears to be a detectable interaction between FN2 and the MukB head domain (MukB-H). Is this due to a contact between N-WHD and MukB-H? It would be useful to see a side-by-side comparison of FN2 interaction with MukB-H and MukB-HN to clarify this.

3) Figure 8/subsection “The MukF N-terminal and C-terminal domains independently modulate MukBEF action in vivo*”* "FN2 overexpression led to a ca. 4-fold more efficient displacement of labelled MukBEF complexes". This comparison is valid only if the overexpression levels and kinetics of FN2 and FC2 are comparable. This should be confirmed.

4) The ATPase assay in Figure 5 lacks data that is referred to in the text, that the Bm2 mutant confers wt ATPase activity.

5) Subsection “The MukF C- and N-terminal domains activate MukB ATPase independently and additively”: The mixture of N- and C-terminal domain of MukF recovered the MukB ATPase activity up to the level in the presence of full-length MukF, leading to a conclusion that the MukB ATPase is activated "additively and independently" by FN and FC. Here, FC is not the MukF C-terminal WHD only, but it additionally includes the preceding linker sequence and the segment binds two MukE molecules (collectively termed MukF middle region). Therefore, the term FC is confusing, and at the same time how MukE binding region could increase the MukB ATPase activity (Figure 7) is surprising. The author generated many MukF fragments. Perhaps, it may be necessary to measure the MukB ATPase activity with other FNs + FCs combinations.

6) Figure 7: First, why was only MukBF normalized? Second, negative control, such as FN2 or FC5 (no MukF middle region) is recommended to be included.

7) If MukB heads display asymmetric ATPase activity as proposed in this manuscript, it should be possible to test this more directly and rigorously using the strong stimulation of MukB ATPase by MukF. For example, does FN stimulate ATP hydrolysis by "free" MukB-head, when bound to a ATP-hydrolysis defective MukB-head-neck construct? Or similarly, does FC bound to an ATP-hydrolysis defective MukB-head stimulate the ATPase activity of "free" MukB heads (which are unable to bind FC)?

8) Figure 4 is taken to suggest that the MukB-F "asymmetric complex contains two independent ATPase domains, one activated by the MukF N-terminal domain and the other activated by the C-terminal domain". The data clearly show that the MukB ATPase can be independently activated by the MukF N- and C-termini. However, it is not at all clear, nor likely, that therefore MukB contains two independent ATPases. In all cases where this was studied, ABC ATPases function as coordinate ATPases. The ATPase may well integrate various levels of independent input, but are still likely to use a concerted reaction to hydrolyse both ATPs. To support a claim that the two ATPases are independent, it is necessary to construct mixed MukB dimers with mutations in one of the two ATPase active sites. Only then can be tested whether the ATPases are indeed independent. Without such data, it is correct to say that both MukF N- and C- termini independently and additively activate the dimeric MukB ATPase.

9) Figure 8: The observed dissolution of MukB foci by overexpression of FN might alternatively be explained by the disruption of MukF dimers by FN. If so, then the presented experiment fails to provide evidence for ring opening at the MukB-MukF-N interface.

10) Figure 7: As control that MukE indeed acts via MukF, please include MukF fragments that stimulate MukB ATPase activity (for example FN2) but do not interact with MukE.

Required textual changes:

The impression given in the text is that MukB-HN differs from MukB-H only in an additional 'neck' region. However, Figure 3 reveals that MukB-HN is over 30 kDa larger than MukB-H. This suggests that MukB-HN contains substantial additional parts of MukB. It is important to give a clear description, at the outset of the experiments, of what these two constructs include.

Introduction, "hawk proteins" are more widely known as HEAT repeat subunits. While you should have the freedom to use one or the other name, we recommend that you spell out the HAWK acronym and that you refer to the presence of HEAT repeats in these proteins.

Subsection “Characterization of the interactions between the MukB neck and MukF”, 'The mutagenesis strategy was informed by MukB and MukF sequence homology and structures […]', this strategy could be better described, e.g. it is not clear in Figure 5 what the Bm1-Bm3 mutations in 'green, blue and orange' are and how they probe interactions of the different faces of the helix. How do they relate to the structures of BsSMC and cohesin, shown in the inserts?

Subsection “Characterization of the interactions between the MukB neck and MukF” discusses the implications of the MukF N-terminal interaction with the MukB neck. "if simultaneous binding of the two necks within the intact MukB dimer by the N-terminal domains of MukF dimer is possible […]". Reviewer 1 thinks that this is an interesting consideration, but that it would initially be more important to describe and discuss how the newly described asymmetric interaction of MukF N- and C-termini with the MukB neck and cup is compatible with the available structural data.

Subsection “MukE inhibits MukBF ATPase” "in agreement with a report by Bahng et al.": the authors should elect to cite other previous studies that have observed inhibition of the MukB ATPase in addition to MukEF.

Subsection “DNA binding to MukB relieves MukE-mediated ATPase inhibition” "The position of the DNA binding interface on MukB heads", please describe where that is.

Figure 8/Subsection “The functional interaction between the MukF N-terminal domain and the MukB neck breaks and reforms during cycles of ATP binding and Hydrolysis” "The analyses reported here demonstrate that the interaction between the MukF N-terminal domain and MukB neck is […] broken and reformed during cycles of ATP binding and hydrolysis." This is a tempting conclusion, but it is not strictly shown by the data. What is shown is that the EQ mutant is resistant to displacement by overexpressed MukF fragments. However, the EQ mutant also shows a very different localisation compared to wild type MukBEF, concentrated at the terminus rather than the origin. The reason for the different response to MukF fragment overexpression could therefore be indirect. It could be either due to the different location in the cell (and whatever the reason for this is), or could be due to the inability to hydrolyse ATP. This point has to be made clear to the readership.

Subsection “The functional interaction between the MukF N-terminal domain and the MukB neck breaks and reforms during cycles of ATP binding and

Hydrolysis” "This result provides further support for a mechanism in which ATP hydrolysis is required to release MukBEF complexes from chromosomes (Nolivos et al., 2016)". Direct biochemical evidence that ATP hydrolysis is required to release an SMC complex from DNA has been provided by Murayama et al., 2015, which is relevant to this context and should be cited accordingly.

Subsection “Perspective” "turnover of cohesion complexes on chromosomes […] is inhibited by acetylation of the SMC subunit, which presumably inhibits SMC3 ATPase". Biochemical evidence that the acetylation acceptor lysines indeed convey DNA-stimulated ATP hydrolysis has also been provided by Murayama et al., 2015, which should therefore be cited.

The Discussion entertains the "rock climber" model for the MukBEF complex, which predicts that MukBEF dimerization mediated by MukF is crucial for the function of the complex. While there is no evidence as yet that other SMC complexes use a similar strategy, it is worth to discuss this interesting proposal in more depth. It could be pointed out that disruption of the MukF dimerization interface, based on the available structural data, seems to be an achievable task that should be able to probe the importance of MukBEF dimerization.

In the Abstract, "turns over" is ambiguous.

Figure 1: ATPγS is invisible.

In Subsection “The MukF N-terminal domain interacts with the MukB neck” and after: Helix 9 is stated to be essential either because it interacts with the neck, or because it is required for proper folding of the helix 8-9. This ambiguous statement is not consistent with the mutagenesis study: targeted mutations on Helix 9 disrupted the interaction of MukF and MukB neck.

Figure 2: There are no FN5 and FN8, and the numbering does not seem to be used with consistence. The naming could be improved.

Figure 2: The label 'resin' is not readily comprehended. 'IgG from resin' would be better.

Subsection “The MukF N-terminal domain interacts with the MukB neck”: "size exclusion chromatography-multi-angle light scattering (SEC-MALS)" has to be in the first paragraph rather than the second paragraph.

Subsection “The MukF N-terminal domain interacts with the MukB neck”, Need to explicitly define FN2 dimer as 2FN.

Subsection “The MukF N-terminal domain interacts with the MukB neck” and on others: The expression like "MukBHN+2FN2 at 1:1.25 monomer:dimer (1:2.5 m:m) molar ratio" is superfluous. "MukBHN+2FN2 at 1:1.25 molar ratio" is succinct and sufficient.

Subsection “The MukF N-terminal domain interacts with the MukB neck”: The statement of "This is consistent with MukBHN, which is a monomer in solution, being unable to form stable heads engaged dimers with either FN2 or FC2 in the presence of ATP." appears in contradiction with the crystal structure of ATP-bound engaged MukB head dimers. Please explain.

Subsection “Characterization of the interactions between the MukB neck and MukF”: The authors should show how the residues on Helix 9 were chosen for mutagenesis: the sequence alignment of kleisins and structural analysis of the Kleisin-Smc neck interactions.

Figure 5: The colors of the structural elements do not match with the Figure legends. Need to label figures (e.g., Helix 8, Helix 9, N-WHD, ScpA, Scc1, MukB, Smc, Smc3). Need to label mutations. In fact, it would be much better to show the entire crystal structure of MukF-MukB, and enlarge the region of mutated positions.

The insets at the bottom of Figure 5 do not help understanding how the mutated residues were selected. As mentioned above, sequence alignment and structural analysis should be shown to clarify this important point. Mutated residues in stick representation should be consistent with Figure 5.

Subsection “MukE inhibits MukBF ATPase”: The sentence, "which cited our previously unpublished observation" is unnecessary.

Subsection “MukE inhibits MukBF ATPase” and Figure 6: Rather than this figure, it is Figure 6—figure supplement 1 that supports the concentration-dependent inhibition of MukF-activated MukB ATPase. In Figure 6 5xDNA is not shown in the table.

Subsection “DNA binding to MukB relieves MukE-mediated ATPase inhibition”: Data have to be shown.

Subsection “The MukF N-terminal and C-terminal domains independently

modulate MukBEF action in vivo*”* and Figure 8: 'FN2' and 'FC2' are stated in the text, but figure shows 'FN2' and 'FC5'.

Subsection “Perspective”: typo, 'held' should be 'hold'. 'cohesion' should be 'cohesin'.

Subsection “Perspective”: "[…], and associated conformational change" would be correct.

Subsection “Perspective”: The last sentence has no verb, and has to be corrected.

Figure 3—figure supplement 2: The legend reads "SEC-MALS analysis of MukBHN-2FN2 and MukBHN-FN2-FC2 complexes in the absence and presence of ATP (1 mM)." but there are no labels on the figure to distinguish which is which. Moreover, this figure appears to be exactly the same as Figure 3—figure supplement 1.

Figure 4—figure supplement 1: There is no graph for (E)!

Figure 5—figure supplement 1: It is surprising to see that only three point mutations induced big changes in the elution volume. Why is this the case? Also, there is no chromatogram for HN+2FN2m1.

Figure 5—figure supplement 4: Panel B is very difficult to grasp and should be improved to make it clearer. At the same time, the authors need to show the model of the full-length MukF dimer interacting with two MukB molecules so that readers could grasp how the MukBEF holocomplex looks like given the newly identified interaction.

Figure 5: Why does MukF not stimulate MukBm1 ATPase activity? It would be expected to bind to MukBm1 via the intact interface and stimulate its activity (as FC does).

The SEC-MALS experiments are very clear. However, a scale for A280 absorption on the y-axis would be helpful.

The choice of residues to be mutated in FN and MukB-neck is poorly described (subsection “Characterization of the interactions between the MukB neck and MukF”). Please include sequence alignments and structural comparisons.

Figure 2: The color-coding is helpful. In addition, the names of the construct should be shown as well.

Subsection “The functional interaction between the MukF N-terminal domain and the MukB neck breaks and reforms during cycles of ATP binding and Hydrolysis”: The manuscript does not directly address (let alone demonstrate) that the interaction is broken and reformed. As written, the claim is too strong and should be softened.

Subsection “MukE and DNA compete during cycles of ATP binding and hydrolysis”. "target to ori" may be better than "relocate to ori".

Based on known structural information: how does a MukF dimer bring together two MukB dimers? Would it be possible to include a tentative model?

---

## [Author Response]

[…] All three reviewers agree that the manuscript provides solid and important new insights into MukBEF biochemistry and that it should appeal to a wide readership. However, the reviewers also raised a number of points, many of which can be addressed through textual changes or changes to the figures, as detailed below. Reviewer 2 found the manuscript very hard to read, and recommends that you work on your figures to make them more comprehensible. The reviewer also would like to see shorter figure legends and a substantially shortened Discussion section. There is also a general feeling that a cartoon drawing of the 'rock climber' model based on the authors' new findings would be desirable.

We have revised the figures substantially to address this point and to make them more accessible to the general reader. We have made figure legends more concise; they now more accurately describe the figures. Furthermore, as requested, we have added an additional figure (Figure 9), which summarizes the findings presented in the paper and shows how these relate to the rock/rope climbing model proposed in outline earlier [which is revised appropriately, taking into account the results here]. We have re-written the Discussion section to make it more concise and incisive.

A strong objection, underlined by all three reviewers (and discussed again at points 7 and 8 of the "Required experimental additions" section), concerns the claim that MukB-F contains two independent ATPase domains. The reviewers believe that the data fail to demonstrate that the architecture of holo-MukBEF complexes is asymmetric. It is possible that a given MukF dimer binds symmetrically to either a single MukB dimer or to a dimer of MukB dimers. It is also unclear, whether FN or FC regulate the two MukB head domains differentially. The presented data are consistent with symmetric ATP hydrolysis by MukB. For instance, binding to FN or to FC might promote MukB head engagement rather than the ATP hydrolysis step. Thus, it seems premature to entertain different timing or different purpose for ATP hydrolysis by the one or the other MukB head domain. It is especially important that, among the other points, you adequately address this specific concern.

We have taken these important concerns into account in our revised manuscript.

Regarding the reviewers’ objection to our suggestion that the independent activation of MukB ATPase hydrolysis by the N- and C-terminal subdomains of MukF may reflect a presence of two independently-controlled ATPases, which become active at different stages of MukBEF-DNA activity cycle, we agree that this was a speculative suggestion. We therefore, have removed any references to this from the Title, Abstract and the Results section, and we have limited any discussion of this to the Discussion section, where it is raised as one possible hypothesis that could explain our data. We discuss this critically in relation to past and emerging new literature (in particular to a new BioRxiv paper from the Rowland-Haering labs; Elbatsh et al., 2017).

With regard to MukB dimers forming an asymmetric complex on ATP (or equivalent) binding in the presence of MukEF, we more explicitly point out that the crystal structure of a MukBEF heads-engaged complex determined by Byung-Ha Oh and colleagues (Woo et al., 2009) demonstrated an asymmetric complex that provides a platform for the interpretation of our data. This structure provided a partial molecular explanation of the asymmetry, because in the heads-engaged state, binding of a second MukF C-terminal domain was sterically prevented by the binding of the MukF middle region attached to the C-terminal domain bound to a head. Furthermore, this paper showed that this displacement of a second MukF C-terminal domain during MukB head engagement occurs in solution and the MukF middle region is required for this displacement. Nevertheless, this work could not address any interaction of the MukF N-terminal domain with the MukB neck, because this domain was absent in the proteins used for crystallography. Our demonstration of such an interaction demonstrates MukBEF likely has an asymmetric architecture comparable to other SMC complexes (notably *B. subtilis* SMC and eukaryote cohesin complexes). We agree that our work does not provide direct evidence for the N-and C-terminal domains of MukF binding to different monomers of MukB dimer and this is clearly pointed out with appropriate schematics in the revised manuscript (Figure 9 and accompanying text). Nevertheless, unless the Woo et al. asymmetric structure is functionally irrelevant, then our demonstration of the MukF-MukB neck interaction, substantiates and strengthens the case for asymmetry in the heads-engaged dimeric complex, although the dimer of dimers complex would be symmetric.

In our opinion, all of this is addressed satisfactorily in the revised manuscript. Parenthetically, if both N- and C-terminal domains within a MukF monomer bind ‘in cis’ to the same MukB monomer, asymmetry would still be present (Figure 9), although there would be less justification for proposing that they activate two different ATPases.

Required experimental additions:1) Figure 2. In this semi-quantitative interaction assay it appears that interaction of the FN2, FN3 and FN4 deletion proteins with MukB is sequentially weakened, in this order. This suggests that the N-WHD and helixes 6 and 7 also make a contribution to MukB interaction. Is this the case? Clarification could come from the inclusion of FN4 in comparison with FN2 in the quantitative interaction assay (shown in Figure 3—figure supplement 3). Else, this should at least be discussed.

The reviewer is correct in pointing out that superficial examination of the semi-quantitative pulldowns, indicates that interaction of N-terminal MukF truncations with MukB or MukBHN may weaken as the variants get smaller (Figure 2 top left and right panels). Nevertheless, closer examination shows that the levels of the respective MukF variants on the FLAG resin also decreases in the same order, consistent with the possibility that the level of pulldown signal is influenced by the amount of ‘bait’ on the resin. Therefore, we see no strong grounds to implicate either the N-WHD or helices 6 and 7 in the interaction. Close examination of the figure makes this point obvious; we now point this out in the revised legend. Importantly, these assays demonstrate that a MukF truncation bearing helices 8 and 9 is sufficient for interaction with the MukB neck in extracts. The variant FN4, containing helices 8 and 9, was not biochemically tractable during and after purification, so we have been unable to study its interaction with the MukB neck by SEC-MALS etc. We now state this in the revised manuscript (Figure 2).

2) In addition, in this experiment, there also appears to be a detectable interaction between FN2 and the MukB head domain (MukB-H). Is this due to a contact between N-WHD and MukB-H? It would be useful to see a side-by-side comparison of FN2 interaction with MukB-H and MukB-HN to clarify this.

Very small amounts of MukBH (slightly above the background level) were reproducibly recovered in samples with FN2 in the pulldowns (note, substantially less pulldown than in combinations in which the interaction was further validated, including interaction of FN2 with MukBHN, as the reviewer requested (Figure 2; top right panel). Nevertheless, we were unable to validate such an interaction by SECMALS, suggesting that either it is not real, or it is insufficiently strong to survive gel filtration. We have now included a sentence about this possible weak contribution to binding in the text (subsection “The MukF N-terminal domain interacts with the MukB neck”).

*3) Figure 8/subsection “The MukF N-terminal and C-terminal domains independently modulate MukBEF action* in vivo*” "FN2 overexpression led to a ca. 4-fold more efficient displacement of labelled MukBEF complexes". This comparison is valid only if the overexpression levels and kinetics of FN2 and FC2 are comparable. This should be confirmed.*

We now include proteins levels (Westerns and quantitative proteomics before and after over-expression in wild type and MukB^EQ^ cells; Figure 8—figure supplement 2). Westerns indicate similar levels of FN2 and FC2 over-expression in MukB^+^ and MukB^EQ^ strains, with >100 excess of these peptides over endogenous MukF levels.

4) The ATPase assay in Figure 5 lacks data that is referred to in the text, that the Bm2 mutant confers wt ATPase activity.

The data showing wild type ATPase activity for mutant Bm2 have been added to Figure 5.

5) Subsection “The MukF C- and N-terminal domains activate MukB ATPase independently and additively”: The mixture of N- and C-terminal domain of MukF recovered the MukB ATPase activity up to the level in the presence of full-length MukF, leading to a conclusion that the MukB ATPase is activated "additively and independently" by FN and FC. Here, FC is not the MukF C-terminal WHD only, but it additionally includes the preceding linker sequence and the segment binds two MukE molecules (collectively termed MukF middle region). Therefore, the term FC is confusing, and at the same time how MukE binding region could increase the MukB ATPase activity (Figure 7) is surprising. The author generated many MukF fragments. Perhaps, it may be necessary to measure the MukB ATPase activity with other FNs + FCs combinations.

Regarding nomenclature, in our opinion, the convention we use is robust, straightforward and explicit. Variants that include the N-terminus of MukF are designated FN, while variants that contain the C-terminus are called FC (the extent of the additional sequences are evident in Figure 2). The only exception to this convention are internal fragments carrying four-helix bundle, which were designated ‘FN’, because they contain the MukF N-terminal domain. We prefer to retain this convention, because other schemes that we could think of were less clear.

The data presented in Figure 7 showing the effect of the middle region on stimulation of MukB ATPase are now discussed more extensively. We are not particularly surprised that increasing the length of the variant led to modest increases in ATPase; this could simply be a consequence increased length enhancing folding or stabilizing the polypeptide. It is not obvious to us how further experiments with other FN + FC combinations would be additionally informative. What is ultimately required is structural information on the interaction!

6) Figure 7: First, why was only MukBF normalized?

Following the reviewer’s comment, we have altered Figure 7; now the bar graph shows all data expressed as absolute ATPase values.

Second, negative control, such as FN2 or FC5 (no MukF middle region) is recommended to be included.

Good point. Now included.

7) If MukB heads display asymmetric ATPase activity as proposed in this manuscript, it should be possible to test this more directly and rigorously using the strong stimulation of MukB ATPase by MukF. For example, does FN stimulate ATP hydrolysis by "free" MukB-head, when bound to a ATP-hydrolysis defective MukB-head-neck construct? Or similarly, does FC bound to an ATP-hydrolysis defective MukB-head stimulate the ATPase activity of "free" MukB heads (which are unable to bind FC)?

See response to 8 below.

8) Figure 4 is taken to suggest that the MukB-F "asymmetric complex contains two independent ATPase domains, one activated by the MukF N-terminal domain and the other activated by the C-terminal domain". The data clearly show that the MukB ATPase can be independently activated by the MukF N- and C-termini. However, it is not at all clear, nor likely, that therefore MukB contains two independent ATPases. In all cases where this was studied, ABC ATPases function as coordinate ATPases. The ATPase may well integrate various levels of independent input, but are still likely to use a concerted reaction to hydrolyse both ATPs. To support a claim that the two ATPases are independent, it is necessary to construct mixed MukB dimers with mutations in one of the two ATPase active sites. Only then can be tested whether the ATPases are indeed independent. Without such data it is correct to say that both MukF N- and C- termini independently and additively activate the dimeric MukB ATPase.

The suggestion of mixed dimers is a good idea, and we have pursued this experimentally for some time. Unfortunately, “free” MukB heads or MukBHNs do not dimerize significantly in the presence of ATP/AMPPNP under any experimental conditions we have tried and did not support significant ATPase levels in the presence or absence of MukF. Therefore, the experiments proposed in 7 above were not informative. Similarly, constructs carrying covalently joined dimeric MukBHNs showed very little MukF-dependent ATPase and are potentially compromised in mutant experiments because a given MukB N-terminal domain can potentially form a ‘head’ with either of the two C-terminal MukB domains in the covalent dimer. Experiments to address this issue are important, but challenging, and in our opinion beyond the scope of this manuscript. Parenthetically and worryingly, we understand from colleagues working with other SMC complexes, that the dimerization hinge may be important for ATPase activity, therefore making the sorts of experiments proposed here even more challenging.

In our opinion, the revised manuscript deals with all of these concerns satisfactorily.

9) Figure 8: The observed dissolution of MukB foci by overexpression of FN might alternatively be explained by the disruption of MukF dimers by FN. If so, then the presented experiment fails to provide evidence for ring opening at the MukB-MukF-N interface.

If this was the case, the foci would also disappear in a MukB^EQ^ mutant strain, unless disruption of dimers was also dependent on cycles of ATP binding and hydrolysis. This is all now discussed in the revised manuscript (subsection “The MukF N-terminal and C-terminal domains independently modulate MukBEF action in vivo*”*).

10) Figure 7: As control that MukE indeed acts via MukF, please include MukF fragments that stimulate MukB ATPase activity (for example FN2) but do not interact with MukE.

Added and addressed above, see point 5.

Required textual changes:The impression given in the text is that MukB-HN differs from MukB-H only in an additional 'neck' region. However, Figure 3 reveals that MukB-HN is over 30 kDa larger than MukB-H. This suggests that MukB-HN contains substantial additional parts of MukB. It is important to give a clear description, at the outset of the experiments, of what these two constructs include.

This has been addressed and corrected at the appropriate positions in the main text and legends.

Introduction, "hawk proteins" are more widely known as HEAT repeat subunits. While you should have the freedom to use one or the other name, we recommend that you spell out the HAWK acronym and that you refer to the presence of HEAT repeats in these proteins.

This has been expanded as requested (Introduction).

Subsection “Characterization of the interactions between the MukB neck and MukF”, 'The mutagenesis strategy was informed by MukB and MukF sequence homology and structures[…]', this strategy could be better described, e.g. it is not clear in Figure 5 what the Bm1-Bm3 mutations in 'green, blue and orange' are and how they probe interactions of the different faces of the helix. How do they relate to the structures of BsSMC and cohesin, shown in the inserts?

To address this comment, we have introduced a new section in Materials and methods section explaining the rationale behind our targeted mutagenesis strategy. We have changed the image of the modelled neck helix to show clearly the mutated residues in Figure 5; and included panel (C) to Figure 5 showing more explicitly the overall similarity of interaction between the kleisins’ helices and the necks in SMCs.

Subsection “Characterization of the interactions between the MukB neck and MukF” discusses the implications of the MukF N-terminal interaction with the MukB neck. "if simultaneous binding of the two necks within the intact MukB dimer by the N-terminal domains of MukF dimer is possible […]". Reviewer 1 thinks that this is an interesting consideration, but that it would initially be more important to describe and discuss how the newly described asymmetric interaction of MukF N- and C-termini with the MukB neck and cup is compatible with the available structural data.

Thank you; this is now discussed in relation to Figure 9 (Discussion section).

Subsection “MukE inhibits MukBF ATPase” "in agreement with a report by Bahng et al.": the authors should elect to cite other previous studies that have observed inhibition of the MukB ATPase in addition to MukEF.

We are not clear what the reviewer is alluding to with regard to inhibition of MukB ATPase. MukB ATPase in the absence of MukEF is negligible. Our searches of PubMed and Google have not revealed previous relevant studies regarding inhibition of MukB(F) ATPase. Nor does the report of Bahng et al. help in providing insight into any other published relevant work.

Subsection “DNA binding to MukB relieves MukE-mediated ATPase inhibition” "The position of the DNA binding interface on MukB heads", please describe where that is.

A Pymol cartoon showing the positions of amino acid residues, whose mutation affected DNA binding in Woo et al., (2009) report has been included in Figure 6.

Figure 8/ Subsection “The functional interaction between the MukF N-terminal domain and the MukB neck breaks and reforms during cycles of ATP binding and Hydrolysis” "The analyses reported here demonstrate that the interaction between the MukF N-terminal domain and MukB neck is […] broken and reformed during cycles of ATP binding and hydrolysis." This is a tempting conclusion, but it is not strictly shown by the data. What is shown is that the EQ mutant is resistant to displacement by overexpressed MukF fragments. However, the EQ mutant also shows a very different localisation compared to wild type MukBEF, concentrated at the terminus rather than the origin. The reason for the different response to MukF fragment overexpression could therefore be indirect. It could be either due to the different location in the cell (and whatever the reason for this is), or could be due to the inability to hydrolyze ATP. This point has to be made clear to the readership.

We agree with the reviewer and this has been addressed in the revised manuscript, by stating an alternative explanation in the revised Results section and by making a ‘softened’ interpretation in the Discussion section.

Subsection “The functional interaction between the MukF N-terminal domain and the MukB neck breaks and reforms during cycles of ATP binding andHydrolysis” "This result provides further support for a mechanism in which ATP hydrolysis is required to release MukBEF complexes from chromosomes (Nolivos et al., 2016)". Direct biochemical evidence that ATP hydrolysis is required to release an SMC complex from DNA has been provided by Murayama et al., 2015, which is relevant to this context and should be cited accordingly.

Now cited.

Subsection “Perspective” "turnover of cohesion complexes on chromosomes […] is inhibited by acetylation of the SMC subunit, which presumably inhibits SMC3 ATPase". Biochemical evidence that the acetylation acceptor lysines indeed convey DNA-stimulated ATP hydrolysis has also been provided by Murayama et al., 2015, which should therefore be cited.

Now cited.

The Discussion entertains the "rock climber" model for the MukBEF complex, which predicts that MukBEF dimerization mediated by MukF is crucial for the function of the complex. While there is no evidence as yet that other SMC complexes use a similar strategy, it is worth to discuss this interesting proposal in more depth. It could be pointed out that disruption of the MukF dimerization interface, based on the available structural data, seems to be an achievable task that should be able to probe the importance of MukBEF dimerization.We agree that this is an excellent idea; experiments testing the functional consequences of disruption of MukF dimerization interface are underway and will be ready for publication soon; this work is not within the scope of the paper here.In the Abstract, "turns over" is ambiguous.

Attended to.

Figure 1: ATPγS is invisible.

Attended to.

In Subsection “The MukF N-terminal domain interacts with the MukB neck” and after: Helix 9 is stated to be essential either because it interacts with the neck, or because it is required for proper folding of the helix 8-9. This ambiguous statement is not consistent with the mutagenesis study: targeted mutations on Helix 9 disrupted the interaction of MukF and MukB neck.

We agree that the original statement was ambiguous and misleading. We have changed the text on in subsection “The MukF N-terminal domain interacts with the MukB neck” (first paragraph) and then when we discuss the point mutants, we state that the requirement for helix 9 is unlikely to be because it is essential for correct folding.

Figure 2: There are no FN5 and FN8, and the numbering does not seem to be used with consistence. The naming could be improved.

Variants were numbered in chronological order as they were constructed. FN5 and FN8, made at the same time as the other variants, are not a subject of the investigation here. Re-numbering all variants to achieve numerical chronology in this manuscript would not facilitate comprehension, but rather would confuse forever the group laboratory note books. We prefer to retain the nomenclature of the original manuscript.

Figure 2: The label 'resin' is not readily comprehended. 'IgG from resin' would be better.

Attended to.

Subsection “The MukF N-terminal domain interacts with the MukB neck”: "size exclusion chromatography-multi-angle light scattering (SEC-MALS)" has to be in the first paragraph rather than the second paragraph.

Attended to.

Subsection “The MukF N-terminal domain interacts with the MukB neck”, Need to explicitly define FN2 dimer as 2FN.

Attended to.

Subsection “The MukF N-terminal domain interacts with the MukB neck” and on others: The expression like "MukBHN+2FN2 at 1:1.25 monomer:dimer (1:2.5 m:m) molar ratio" is superfluous. "MukBHN+2FN2 at 1:1.25 molar ratio" is succinct and sufficient.

We agree; corrected.

Subsection “The MukF N-terminal domain interacts with the MukB neck”: The statement of "This is consistent with MukBHN, which is a monomer in solution, being unable to form stable heads engaged dimers with either FN2 or FC2 in the presence of ATP." appears in contradiction with the crystal structure of ATP-bound engaged MukB head dimers. Please explain.

In our opinion, there is no contradiction. The crystal structure was made in the presence of MukE. In addition, high concentration of proteins in samples prepared for crystal structure experiments and crystal packing will have stabilized the ‘heads engaged’ complex.

Subsection “Characterization of the interactions between the MukB neck and MukF”: The authors should show how the residues on Helix 9 were chosen for mutagenesis: the sequence alignment of kleisins and structural analysis of the Kleisin-Smc neck interactions.

The sequence conservation between MukBEF and other kleisins-SMCs is poor. The rationale for targeted mutagenesis of helix9-neck interface is now provided as a separate section in Material and Methods. We have now extended analysis of the impact of mutations in helix 9 by assaying the ability of MukF variants carrying these mutations to suppress the effects of MukF deletion gene in in vivo complementation assays (Figure 5—figure supplement 2).

Figure 5: The colors of the structural elements do not match with the Figure legends. Need to label figures (e.g., Helix 8, Helix 9, N-WHD, ScpA, Scc1, MukB, Smc, Smc3). Need to label mutations. In fact, it would be much better to show the entire crystal structure of MukF-MukB, and enlarge the region of mutated positions.The insets at the bottom of Figure 5 do not help understanding how the mutated residues were selected. As mentioned above, sequence alignment and structural analysis should be shown to clarify this important point. Mutated residues in stick representation should be consistent with Figure 5.

We have revised Figure 5 to make all of the above clear. The published crystal structure of MukBEF (Woo et al., 2009) does not include the region of the neck, or the MukF N-terminal domain.

Subsection “MukE inhibits MukBF ATPase”: The sentence, "which cited our previously unpublished observation" is unnecessary.

Revised as proposed.

Subsection “MukE inhibits MukBF ATPase” and Figure 6: Rather than this figure, it is Figure 6—figure supplement 1 that supports the concentration-dependent inhibition of MukF-activated MukB ATPase.

Attended to (subsection “MukE inhibits MukBF ATPase”)

In Figure 6 5xDNA is not shown in the table.

Now shown.

Subsection “DNA binding to MukB relieves MukE-mediated ATPase inhibition”: Data has to be shown.

Data is now shown in the revised manuscript; subsection “DNA binding to MukB relieves MukE-mediated ATPase inhibition”.

Subsection “The MukF N-terminal and C-terminal domains independently

*modulate MukBEF action* in vivo*” and Figure 8: 'FN2' and 'FC2' are stated in the text, but figure shows 'FN2' and 'FC5'.*

The reviewer is correct, FC5 has been used in the experiment and this is now corrected in the text.

Subsection “Perspective”: typo, 'held' should be 'hold'. 'cohesion' should be 'cohesin'.

Yes, thank you, we changed the first typo; ‘cohesion’ is correct.

Subsection “Perspective”: "[…], and associated conformational change" would be correct.

Yes, thank you. We have made the correction.

Subsection “Perspective”: The last sentence has no verb, and has to be corrected.

Corrected. Thank you.

Figure 3—figure supplement 2: The legend reads "SEC-MALS analysis of MukBHN-2FN2 and MukBHN-FN2-FC2 complexes in the absence and presence of ATP (1 mM)." but there are no labels on the figure to distinguish which is which. Moreover, this figure appears to be exactly the same as Figure 3—figure supplement 1.

OOOPs! The revised manuscript corrects this. In the initial submission, Figure 3—figure supplement 1 was placed twice, firstly labelled correctly and second as Figure 3—figure supplement 2. The correct Figure 3—figure supplement 2 is present and all labels are now correct.

Figure 4—figure supplement 1: There is no graph for (E)!

Sorry-there should have been no reference to panel (E) in the legend and that part of the legend has been deleted. There is no reference to this in the text. We understand also that the reviewer wishes to see the titrations in the presence of MukE; these are now present in Figure 6—figure supplement 1.

Figure 5—figure supplement 1: It is surprising to see that only three point mutations induced big changes in the elution volume. Why is this the case? Also, there is no chromatogram for HN+2FN2m1.

The initial figure may have confused the reviewer-sorry! We do not know why the same proteins (whether it be 2FN2 or MukB_HN_) do not elute at quite the same elution volume in different runs. Furthermore, our initial figure labelling was not clear and may have confused the reviewer. This has been attended to in the revised manuscript. What is clear from the profiles is that 2FN2-MukB_HN_ complexes form when wild type 2FN2 is present, but not when the 2FN2 mutants are used. In our opinion, there is no ambiguity to this conclusion in the revised figure.

Figure 5—figure supplement 4: Panel B is very difficult to grasp and should be improved to make it clearer. At the same time, the authors need to show the model of the full-length MukF dimer interacting with two MukB molecules so that readers could grasp how the MukBEF holocomplex looks like given the newly identified interaction.

The model is now presented in Figure 9. Additionally, Figure 5 is improved.

Figure 5: Why does MukF not stimulate MukBm1 ATPase activity? It would be expected to bind to MukBm1 via the intact interface and stimulate its activity (as FC does).

We do not know why this mutant in the MukB neck does not have its ATPase activated by interaction of the MukF C-terminal domain. We do have evidence that this protein is biochemically active as it binds to MukFE, but further work, outside of the scope of this manuscript, would be needed to investigate the reasons for failed activation by FC2. Because of this somewhat puzzling result, we have now analysed MukBm3 further. This variant had ~35% of ATPase activity in the presence of MukF (Figure 5). Importantly, we now show that FC2 activated MukBm3 ATPase as efficiently as MukF, while, FN2 failed to show any activation (Figure 5—figure supplement 3). We now include these new data and an appropriate comment on the puzzling properties of MukBm1.

The SEC-MALS experiments are very clear. However, a scale for A280 absorption on the y-axis would be helpful.

The scale is now included in all of the SEC-MALS experiments shown.

The choice of residues to be mutated in FN and MukB-neck is poorly described (subsection “Characterization of the interactions between the MukB neck and MukF”). Please include sequence alignments and structural comparisons.

The sequence conservation between MukBEF and other kleisins-SMCs is

poor. The rationale for targeted mutagenesis of the helix 9-neck interface is

now provided explicitly as a separate section in Material and Methods.

Figure 2: The color-coding is helpful. In addition, the names of the construct should be shown as well.

Attended to.

Subsection “The functional interaction between the MukF N-terminal domain and the MukB neck breaks and reforms during cycles of ATP binding and Hydrolysis”: The manuscript does not directly address (let alone demonstrate) that the interaction is broken and reformed. As written, the claim is too strong and should be softened.

The revised manuscript addresses this point both in the Results and Discussion sections.

Subsection “MukE and DNA compete during cycles of ATP binding and hydrolysis”. "target to ori" may be better than "relocate to ori".

We have changed the text to “target to ori” as proposed.

Based on known structural information: how does a MukF dimer bring together two MukB dimers? Would it be possible to include a tentative model?

Schematics and a model are now included in Figure 9; the model extends a schematic initially published in Badrinarayanan et al., 2012.